# On the Scalability and Memory Efficiency of Semidefinite Programs for Lipschitz Constant Estimation of Neural Networks: Scaling the Computation for ImageNet

**Zi Wang**[*1], **Bin Hu**[*2], **Aaron J Havens**[2], **Alexandre Araujo**[3], **Yang Zheng**[4], **Yudong Chen**[1], **Somesh Jha**[1]
1. Department of Computer Sciences, University of Wisconsin-Madison
2. CSL & ECE, University of Illinois Urbana-Champaign
3. ECE, New York University
4. ECE, University of California San Diego

## Abstract

Lipschitz constant estimation plays an important role in understanding generalization, robustness, and fairness in deep learning. Unlike naive bounds based on the network weight norm product, semidefinite programs (SDPs) have shown great promise in providing less conservative Lipschitz bounds with polynomial-time complexity guarantees. However, due to the memory consumption and running speed, standard SDP algorithms cannot scale to modern neural network architectures. In this paper, we transform the SDPs for Lipschitz constant estimation into an eigenvalue optimization problem, which aligns with the modern large-scale optimization paradigms based on first-order methods. This is amenable to autodiff frameworks such as PyTorch and TensorFlow, requiring significantly less memory than standard SDP algorithms. The transformation also allows us to leverage various existing numerical techniques for eigenvalue optimization, opening the way for further memory improvement and computational speedup. The essential technique of our eigenvalue-problem transformation is to introduce redundant quadratic constraints and then utilize both Lagrangian and Shor's SDP relaxations under a certain trace constraint. Notably, our numerical study successfully scales the SDP-based Lipschitz constant estimation to address large neural networks on ImageNet. Our numerical examples on CIFAR10 and ImageNet demonstrate that our technique is more scalable than existing approaches. Our code is available at `https://github.com/z1w/LipDiff`.

## 1 Introduction

The Lipschitz constant of a neural network plays an important role in training stability (Miyato et al., 2018), robustness (Tsuzuku et al., 2018), and generalization (Bartlett et al., 2017). Given the fact that computing the Lipschitz constant of a neural network is NP-hard (Scaman & Virmaux, 2018), researchers have attempted to improve upon the naive norm product bound and devise methods to compute tighter and more efficient upper bounds to approximate this Lipschitz constant.

Recently, researchers have proposed to use semidefinite programs (SDPs) to estimate the Lipschitz constant of neural networks (Fazlyab et al., 2019; Wang et al., 2022). LipSDP (Fazlyab et al., 2019) is among the first SDP-based conditions that are used to estimate the Lipschitz constant of neural networks and has motivated numerous works on designing smooth network structures for achieving improved certified robustness (Araujo et al., 2023a; Wang & Manchester, 2023; Havens et al., 2023; Fazlyab et al., 2024). Wang et al. (2022) have drawn the connections between the Lipschitz constant estimation problem and the matrix mixed norm problems, and hinted that semidefinite programming

---

*  * Equal contribution.

is likely the optimal method to estimate the Lipschitz constant within polynomial time assuming standard complexity theoretical conjectures.

Though SDPs are convex programs and can be solved by interior point methods (IPMs) within polynomial time (Ben-Tal & Nemirovski, 2001), the memory requirement of IPMs is enormous. Consequently, LipSDP cannot scale to modern neural network architectures, which contain at least thousands of neurons. Currently, general-purpose SDP solvers are not capable of addressing the scalability issue of LipSDP, placing a significant hurdle for practical use of such methods. Recently, researchers have studied how to improve the scalability of LipSDP (Xue et al., 2022; Newton & Papachristodoulou, 2021). These methods mainly exploit the chordal sparsity contained in SDPs (Zheng et al., 2021). Chordal sparsity allows us to decompose a large SDP constraint to a set of smaller and coupled ones which might be more scalable to solve; see Zheng et al. (2021) for a recent review. However, these methods still cannot scale to practical networks on CIFAR10 or ImageNet.

In this work, we aim to address the memory efficiency and scalability of LipSDP. Instead of exploiting chordal sparsity, we transform the SDP-based Lipschitz constant estimation into an eigenvalue optimization problem using exact penalty methods (Ruszczynski, 2011) and redundant trace constraint arguments (Liao et al., 2023). Our nonsmooth eigenvalue perspective is motivated by Dathathri et al. (2020) as well as the advances in large-scale SDP optimization using spectral bundle methods (Helmberg & Rendl, 2000; Ding & Grimmer, 2023; Liao et al., 2023). In addition to the eigenvalue problem transformation, we also provide several add-on techniques that enable faster eigenvalue estimation for SDPs induced from large networks and provide more effective strategies for initializing first-order subgradient methods. Previously, for large networks, we can only estimate the Lipschitz constant using the spectral norm product of weight matrices (Leino et al., 2021). With our initialization, we can recover this bound from the beginning of the iterative optimization process. In other words, we transform the Lipschitz constant estimation into an incremental optimization process, which is guaranteed to improve the spectral norm product and is convenient to terminate at any time. Our key contributions are summarized as follows.

- We introduce an eigenvalue optimization formulation for LipSDP, which we call EP-LipSDP. Based on exact penalty methods, we add redundant trace constraints to prove that EP-LipSDP is equivalent to LipSDP. We obtain the exact penalty parameter by properly introducing redundant quadratic constraints. EP-LipSDP has a special form such that first-order subgradient methods can be directly applied.
- We propose a series of numerical techniques compatible with EP-LipSDP, such as eigenvector approximation, sparse matrix multiplication, and analytical initialization. These techniques further improve the speed and practical convergence of solving EP-LipSDP.
- We implemented the algorithm as **LipDiff** and evaluated it on practical neural networks on CIFAR10 and ImageNet. Notably, our numerical study successfully scales the SDP-based Lipschitz constant estimation to address large networks on ImageNet. Our numerical examples on CIFAR10 and ImageNet demonstrate that EP-LipSDP is more scalable than existing approaches.

## 2   RELATED WORK

The Lipchitz constant estimation of neural networks has been extensively studied in many recent works (Scaman & Virmaux, 2018; Fazlyab et al., 2019; Latorre et al., 2020; Jordan & Dimakis, 2020; Wang et al., 2022; Chen et al., 2020; 2021; Pauli et al., 2024). This problem is in general NP-hard (Scaman & Virmaux, 2018; Jordan & Dimakis, 2020; Wang et al., 2022), and thus an exact estimation is infeasible in practice. It is more plausible to leverage approximations in convex forms such as LipSDP (Fazlyab et al., 2019; Wang et al., 2022). However, a main issue for LipSDP is the lack of scalability to large networks. Recently, researchers have leveraged chordal sparsity to improve the scalability of LipSDP (Xue et al., 2022; Newton & Papachristodoulou, 2021; Batten et al., 2021). Noticeably, such methods still cannot scale to networks on ImageNet or CIFAR10.

Interestingly, Dathathri et al. (2020) transformed the SDPs designed for neural-network local robustness verification (which is a different problem from Lipschitz constant estimation) into an eigenvalue problem that can be solved using subgradient methods. Our approach is inspired by Dathathri et al. (2020), and focuses on a different problem, namely Lipschitz constant estimation. We develop new optimization techniques tailored specifically for Lipschitz constant estimation, which enables anytime termination with a quality guarantee.

It is worth mentioning that there have been significant research efforts on improving the scalability of general SDPs via first-order methods (Wen et al., 2010; O'donoghue et al., 2016; Zheng et al., 2020; Garstka et al., 2021); see Majumdar et al. (2020) and Zheng et al. (2021) for recent surveys. The formulation in this paper closely follows the nonsmooth eigenvalue perspective pioneered by Helmberg & Rendl (2000). This nonsmooth perceptive is well-suited for applying first-order methods, from vanilla subgradient methods and cutting plane techniques to sophisticated bundle methods (Lemarechal & Zowe, 1994). Helmberg & Rendl (2000) developed a special class of spectral bundle methods for solving the nonsmooth eigenvalue formulation of dual SDPs with constant trace constraints. Further developments appeared in Helmberg & Kiwiel (2002) and Helmberg et al. (2014). The constant trace property has also been investigated in polynomial optimization literature (Mai et al., 2022). Very recently, comprehensive convergence rates of spectral bundle methods have been established in Ding & Grimmer (2023), and a comparison of spectral bundle methods for primal and dual SDPs is discussed in Liao et al. (2023).

Built upon LipSDP (Fazlyab et al., 2019), several works (Araujo et al., 2023b; Wang & Manchester, 2023; Havens et al., 2023) have also developed direct parameterizations of SDP-based Lipschitz network layers. This allows the design of specific network structures with prescribed Lipschitz guarantees, leading to non-trivial certified robust accuracy. It is possible that our proposed analysis method can be used to develop less conservative Lipschitz network structures in the future.

## 3  BACKGROUND

**Notation**. We denote the $n \times n$ identity matrix and the $n \times n$ zero matrix as $I_{n \times n}$ and $0_{n \times n}$, respectively. We will omit the subscripts when the dimension is clear from the context. Given a vector $v$, the notation $v \geq 0$ means that all the entries of $v$ are non-negative. We denote the $i$-th entry of $v$ as $v_i$. In addition, we use the notation $\mathrm{diag}(v)$ to denote the diagonal matrix whose $(i, i)$-th entry is equal to $v_i$. For a vector $v$, $\|v\|$ denotes the Euclidean norm of $v$, i.e., $\|v\| = \sqrt{\sum v_i^2}$. For a matrix $M$, we use $\|M\|_{\mathrm{op}}$ to denote the operator norm, i.e., the largest singular value of $M$.

### 3.1  A BRIEF REVIEW OF LIPSDP

The LipSDP approach can be formulated in either the primal or dual domain, leading to a pair of SDP conditions for the Lipschitz constant estimation of general neural networks. The original LipSDP was developed in the dual domain (Fazlyab et al., 2019). The primal form of LipSDP was later discussed in Wang et al. (2022; 2023). To illustrate different forms of LipSDP, we first consider the simplest single-layer neural network in the form of $f(x) = v\sigma(Wx + b^0) + b^1$, where $x \in \mathbb{R}^{n_x}$, $W \in \mathbb{R}^{n \times n_x}$, $b^0 \in \mathbb{R}^n$, $v \in \mathbb{R}^{1 \times n}$, $b^1 \in \mathbb{R}$, and $f(x) \in \mathbb{R}$. Here $f$ is a scalar, and $n$ is the neuron number. The activation $\sigma$ is assumed to be slope-restricted on $[0, 1]$. In the dual domain, the LipSDP (Fazlyab et al., 2019) is formulated as follows:

$$\min_{\zeta, \tau} \quad \zeta$$
$$\text{subject to} \quad \begin{bmatrix} -\zeta I & W^\mathsf{T} \mathrm{diag}(\tau) \\ \mathrm{diag}(\tau) W & -2\,\mathrm{diag}(\tau) + v^\mathsf{T} v \end{bmatrix} \preceq 0, \quad \tau \geq 0, \tag{1}$$

where $\tau \in \mathbb{R}^n$ and $\zeta \in \mathbb{R}$ are the decision variables. The original proof in Fazlyab et al. (2019) is based on the quadratic constraint argument. Given two arbitrary $x, x' \in \mathbb{R}^{n_x}$, set $z = \sigma(Wx + b^0)$ and $z' = \sigma(Wx' + b^0)$. Denote $\Delta x = x' - x$ and $\Delta z = z' - z$. If the matrix inequality (1) holds, we must have

$$\begin{bmatrix} \Delta x \\ \Delta z \end{bmatrix}^\mathsf{T} \begin{bmatrix} -\zeta I & W^\mathsf{T} \mathrm{diag}(\tau) \\ \mathrm{diag}(\tau) W & -2\,\mathrm{diag}(\tau) + v^\mathsf{T} v \end{bmatrix} \begin{bmatrix} \Delta x \\ \Delta z \end{bmatrix} \leq 0$$

which is equivalent to

$$\|\Delta z\|^2 - \zeta \|\Delta x\|^2 + \begin{bmatrix} \Delta x \\ \Delta z \end{bmatrix}^\mathsf{T} \begin{bmatrix} 0 & W^\mathsf{T} \mathrm{diag}(\tau) \\ \mathrm{diag}(\tau) W & -2\,\mathrm{diag}(\tau) \end{bmatrix} \begin{bmatrix} \Delta x \\ \Delta z \end{bmatrix} \leq 0.$$

The third term on the left side is non-negative since $\sigma$ is slope-restricted on $[0, 1]$ (see Fazlyab et al. (2019, section 2.2) and Wang et al. (2022, section 4) for detailed explanations). Then the sum of

the first two terms has to be non-positive, and one has $\|\Delta z\| \leq \sqrt{\zeta} \|\Delta x\|$. Hence one just needs to minimize $\zeta$ subject to the matrix inequality (1).

In the primal side, we let $W_i$ denote the $i$-th row of $W$, and one starts with the following problem

$$\max \quad \frac{|v\Delta z|}{\|\Delta x\|}$$
$$\text{subject to} \quad (\Delta z_i - W_i \Delta x)\Delta z_i \leq 0, \ \forall i \in [n],$$

whose solution gives an upper bound for the Lipschitz constant of $f$ due to the fact that $\sigma$ is slope-restricted on $[0, 1]$. Since scaling $\Delta x$ and $\Delta z$ by the same constant always maintains the constraint, one can show that the solution for the above problem is further upper bounded by the optimal value of the following quadratically constrained quadratic program (QCQP):

$$\max \quad v\Delta z$$
$$\text{subject to} \quad (\Delta z_i - W_i \Delta x)\Delta z_i \leq 0, \ \forall i \in [n] \tag{2}$$
$$\|\Delta x\| \leq 1.$$

Then Shor's relaxation of the above QCQP gives the primal form of LipSDP, which is the dual SDP of (1). Specifically, the dual program of Shor's relaxation of (2) is the following SDP:

$$\min_{\zeta, \tau, \gamma} \quad \zeta$$
$$\text{subject to} \quad \begin{bmatrix} \gamma - \zeta & 0 & v \\ 0 & -\gamma I & W^\mathsf{T} \operatorname{diag}(\tau) \\ v^\mathsf{T} & \operatorname{diag}(\tau)W & -2\operatorname{diag}(\tau) \end{bmatrix} \preceq 0, \tag{3}$$
$$\zeta \geq 0, \ \tau \geq 0, \ \gamma \geq 0.$$

which can be shown to give the same Lipschitz upper bound as (1) via Schur complement and some rescaling arguments[1]

The primal and dual forms of LipSDP give the same solution. One can also modify the above arguments to connect the primal and dual LipSDP forms for the more general case where $f$ is a multi-layer network with a vector output. In this paper, we will show that the solution to the original LipSDP for multi-layer networks (Fazlyab et al., 2019, Theorem 2) can be recovered by another optimization problem that can be efficiently solved using first-order methods.

## 4 MAIN THEORETICAL RESULTS

The semidefinite constraint in LipSDP is non-trivial to deal with for large-scale problems. One common idea for improving scalability and memory-efficiency is to move the semidefinite constraint into the cost function via the exact penalty method (Ruszczynski, 2011). In this section, we derive the exact penalty form of LipSDP (termed as EP-LipSDP). We show that EP-LipSDP and LipSDP have the same solutions. In addition, EP-LipSDP is an optimization problem with only box constraints and hence can be efficiently solved using first-order methods. To illustrate the essence of our technique, we first present EP-LipSDP for the single-layer setting reviewed in Section 3.1. After that, we present EP-LipSDP for the multi-layer network case.

### 4.1 EP-LIPSDP FOR SINGLE-LAYER NEURAL NETWORK

Recall that we have $f(x) = v\sigma(Wx + b^0) + b^1$, where $x \in \mathbb{R}^{n_x}$, $W \in \mathbb{R}^{n \times n_x}$, $b^0 \in \mathbb{R}^n$, $v \in \mathbb{R}^{1 \times n}$, $b^1 \in \mathbb{R}$, and $f(x) \in \mathbb{R}$. The neural network $f : \mathbb{R}^{n_x} \to \mathbb{R}$, has $n$ neurons, and $\sigma$ is assumed to be slope-restricted on $[0, 1]$. For now, we consider the output dimension of $f$ to be 1. Due to the page limit, the discussion on the vector output case is deferred to Appendix F. A naive way to transform LipSDP to a penalty form is to move the semidefinite constraint in (1) to the cost:

$$\min_{\zeta \geq 0, \lambda \geq 0} \quad \zeta + \rho \lambda_{\max}^+ \left( \begin{bmatrix} -\zeta I & W^\mathsf{T} \operatorname{diag}(\tau) \\ \operatorname{diag}(\tau)W & -2\operatorname{diag}(\tau) + v^\mathsf{T} v \end{bmatrix} \right), \tag{4}$$

---

[1]Notice that the network $f$ is $\frac{\zeta}{2}$-Lipschitz if the semidefinite constraint in (3) is feasible. Hence the optimal value of (3) corresponds to the product of 2 and the square root of the optimal value of (1). For completeness, we include a detailed proof in Appendix E. Some related arguments are also given in Wang et al. (2022; 2023).

where $\lambda_{\max}^+(\cdot) = \max(0, \lambda_{\max}(\cdot))$ with $\lambda_{\max}(\cdot)$ denoting the maximum eigenvalue, and $\rho > 0$ is a penalty parameter (Ruszczynski, 2011, Theorem 7.21). One can then apply first-order methods to solve this new nonsmooth optimization problem (4). However, a crucial issue is that it is unclear how large the penalty parameter $\rho$ has to be such that (4) and (1) can yield the same solution. The key issue here is that the dual form of (1) (which is the Shor's SDP relaxation of (2)) does not have an explicit trace constraint. Based on the discussion in (Liao et al., 2023, Section 3.2), redundant trace constraints may be used to derive the exact penalty form of SDPs. Next, we will enforce a redundant trace constraint on the dual program of (3) to derive the exact penalty form of LipSDP.

We need to introduce an explicit trace constraint to the dual program of (1) without affecting its solution. We start by adding redundant constraints into (2) such that its Shor's SDP relaxation has an explicit trace constraint. Specifically, the constraint in (2) already states $\|\Delta x\| \le 1$. Augmenting the inequality $(\Delta z_i - W_i \Delta x)\Delta z_i \le 0$ leads to a redundant constraint $\Delta z_i^2 \le \|W_i \Delta x\|^2 \le \|W_i\|_{\mathrm{op}}^2$ (one interpretation for this redundant constraint is that $\sigma$ is slope-restricted on $[0,1]$ and hence has to be 1-Lipschitz). Therefore, the following QCQP is feasible if and only if (2) is feasible:

$$
\begin{aligned}
\max \quad & v\Delta z \\
\text{subject to} \quad & (\Delta z_i - W_i \Delta x)\Delta z_i \le 0, \ \forall i \in [n] \\
& \|\Delta x\|_2 \le 1 \\
& \Delta z_i^2 \le \|W_i\|_{\mathrm{op}}^2, \ \forall i \in [n].
\end{aligned}
\tag{5}
$$

Denote $a_i = \|W_i\|_{\mathrm{op}}^2$. Then Shor's SDP relaxation of (5) has an explicit trace constraint $1 + \|\Delta x\|^2 + \|\Delta z\|^2 \le 2 + \sum_{i=1}^n a_j$. One can verify that the dual of the Shor's SDP relaxation of (5) reads as

$$
\begin{aligned}
\min_{\zeta, \lambda, \tau, \gamma} \quad & \zeta \\
\text{subject to} \quad & \begin{bmatrix} \sum_{j=1}^n a_j \lambda_j + \gamma - \zeta & 0 & v \\ 0 & -\gamma I & W^\mathsf{T} \operatorname{diag}(\tau) \\ v^\mathsf{T} & \operatorname{diag}(\tau)W & -2\operatorname{diag}(\tau) - \operatorname{diag}(\lambda) \end{bmatrix} \preceq 0, \\
& \zeta \ge 0, \ \lambda \ge 0, \ \tau \ge 0, \ \gamma \ge 0.
\end{aligned}
\tag{6}
$$

Here $\tau \in \mathbb{R}^n$ corresponds to the slope-restricted property, $\lambda \in \mathbb{R}^n$ corresponds to the redundant constraints $\Delta z_i^2 \le \|W_i\|_2^2$, and $\gamma$ corresponds to the constraint $\|\Delta x\|_2 \le 1$. Comparing (6) with (3), the only difference is that the redundant variable $\lambda$ shows up in the $(1,1)$-block and $(3,3)$-block. Intuitively, since the dual program of (6) has an explicit trace constraint, the penalty formulation of (6) will be exact; see (Liao et al., 2023, Section 3.2) for discussions. For simplicity, we define

$$
C := \begin{bmatrix} \sum_{j=1}^n a_j \lambda_j + \gamma - \zeta & 0 & v \\ 0 & -\gamma I & W^\mathsf{T} \operatorname{diag}(\tau) \\ v^\mathsf{T} & \operatorname{diag}(\tau)W & -2\operatorname{diag}(\tau) - \operatorname{diag}(\lambda) \end{bmatrix}.
$$

Now we can transform (6) to its exact penalty form, leading to the following EP-LipSDP:

$$
\min_{\zeta \ge 0, \lambda \ge 0, \tau \ge 0, \gamma \ge 0} \ \zeta + \left(2 + \sum_{j=1}^n a_j\right)\lambda_{\max}^+(C).
\tag{7}
$$

The penalty parameter is set as $2 + \sum_{j=1}^n a_j$, which is the trace bound in the dual SDP of (6). The dual programs of (6) and (3) differ by only a redundant trace constraint. Hence EP-LipSDP (7) and the original LipSDP (1) give the same Lipschitz bounds. This leads to the following main result.

**Theorem 1.** *Let $\mathbf{opt}_1$ be the optimal value of Problem (6) and $\mathbf{opt}_2$ be the optimal value of Problem (7). In addition, let $\mathbf{opt}_3$ be the optimal value of the original LipSDP (1). We have $\mathbf{opt}_1 = \mathbf{opt}_2 = 2\sqrt{\mathbf{opt}_3}$. Therefore, all three programs give the same Lipschitz bounds.*

The detailed proof of the above theorem is presented in Appendix A. The nonsmooth problem (7) can be solved via subgradient methods and bundle methods (Helmberg & Rendl, 2000; Ding & Grimmer, 2023; Liao et al., 2023). In our case of Lipschitz constant estimation, we can utilize autodiff frameworks such as PyTorch to get (sub)gradients of the cost function (7). Thus, this eigenvalue transformation opens the door to applying modern machine learning architectures and first-order techniques based on sparse eigen-solvers and iterative subgradient methods. Finally, if we increase the penalty parameter such that $\rho > 2 + \sum_{j=1}^n a_j$, then we can ensure that the solutions (not just the optimal values) of (6) and (7) are equivalent. See Appendix A for a detailed discussion.

## 4.2 EP-LipSDP for Multi-layer Neural networks

Next, we present the result for the general multi-layer case. For ease of exposition, we still consider the scalar output case. The extension to the vector output is explained in Appendix F. Specifically, consider a multi-layer network:

$$x^{(0)} = x, \quad x^{(k)} = \phi(W^{(k-1)}x^{(k-1)} + b^{(k-1)}) \quad k = 1, \ldots, d, \quad f(x) = vx^{(d)} + b^{(d)}. \quad (8)$$

Suppose $x_k \in \mathbb{R}^{n_k}$. Again, we consider the case that $v$ is a row vector and $f$ is a scalar output. The original LipSDP has the following form:

$$\min_{\zeta, \{\tau^{(k)}\}_{k=1}^d} \zeta$$

$$\text{subject to} \quad \Lambda_k = \text{diag}(\tau^{(k)}), \quad \tau^{(k)} \geq 0, \quad \tau^{(k)} \in \mathbb{R}^{n_k}, \quad \zeta \geq 0, \quad \zeta \in \mathbb{R},$$

$$\begin{bmatrix} -\zeta I & (W^{(0)})^\mathsf{T}\Lambda_1 & 0 & \cdots & & 0 \\ \Lambda_1 W^{(0)} & -2\Lambda_1 & \ddots & \ddots & & \vdots \\ 0 & \ddots & \ddots & (W^{(d-2)})^\mathsf{T}\Lambda_{d-1} & & 0 \\ \vdots & \ddots & \Lambda_{d-1}W^{(d-2)} & -2\Lambda_{d-1} & (W^{(d-1)})^\mathsf{T}\Lambda_d \\ 0 & 0 & 0 & \Lambda_d W^{(d-1)} & -2\Lambda_d + v^\mathsf{T}v \end{bmatrix} \preceq 0.$$
$$(9)$$

Similar to the equivalent relationship between (1) and (3), we can show that the following SDP can give the same Lipschitz bound to (9):

$$\min_{\zeta, \gamma, \{\tau^{(k)}\}_{k=1}^d} \zeta$$

$$\text{subject to} \quad \zeta \geq 0, \quad \gamma \geq 0, \quad \Lambda_k = \text{diag}(\tau^{(k)}), \quad \tau^{(k)} \geq 0, \quad \tau^{(k)} \in \mathbb{R}^{n_k}, \text{for} \quad k = 1, \cdots, d$$

$$\begin{bmatrix} \gamma - \zeta & 0 & 0 & 0 & \cdots & v \\ 0 & -\gamma I & (W^{(0)})^\mathsf{T}\Lambda_1 & 0 & \cdots & 0 \\ 0 & \Lambda_1 W^{(0)} & -2\Lambda_1 & \ddots & \ddots & \vdots \\ 0 & 0 & \ddots & \ddots & (W^{(d-2)})^\mathsf{T}\Lambda_{d-1} & 0 \\ \vdots & \vdots & \ddots & \Lambda_{d-1}W^{(d-2)} & -2\Lambda_{d-1} & (W^{(d-1)})^\mathsf{T}\Lambda_d \\ v^\mathsf{T} & 0 & 0 & 0 & \Lambda_d W^{(d-1)} & -2\Lambda_d \end{bmatrix} \preceq 0.$$
$$(10)$$

Notice (10) can be viewed as the multi-layer extension of (3). Denote the $i$-th row of $W^k$ as $W_i^{(k)}$. For $k = 1, \cdots, d$, denote $c_{kj} = \left\|W_j^{(k-1)}\right\|_{\text{op}}^2 \cdot \left(\prod_{i=0}^{k-2} \left\|W^{(i)}\right\|_{\text{op}}^2\right)$ for $j = 1, \cdots, n_k$. Then we can use the same argument for the single-layer case to show that $2 + \sum_{k=1}^d \sum_{j=1}^{n_k} c_{kj}$ provides an explicit redundant trace constraint to the multi-layer QCQP, and hence can be used to derive the exact penalty formulation of the multi-layer LipSDP. Specifically, we define $S_k = \text{diag}(\lambda^{(k)})$ with $\lambda^{(k)} \geq 0$ being a vector in $\mathbb{R}^{n_k}$. Then we define

$$C = \begin{bmatrix} \sum_{k,j} c_{kj}\lambda_j^{(k)} + \gamma - \zeta & 0 & 0 & 0 & \cdots & v \\ 0 & -\gamma I & (W^{(0)})^\mathsf{T}\Lambda_1 & 0 & \cdots & 0 \\ 0 & \Lambda_1 W^{(0)} & -2\Lambda_1 - S_1 & \ddots & \ddots & \vdots \\ 0 & 0 & \ddots & \ddots & (W^{(d-2)})^\mathsf{T}\Lambda_{d-1} & 0 \\ \vdots & \vdots & \ddots & \Lambda_{d-1}W^{(d-2)} & -2\Lambda_{d-1} - S^{d-1} & (W^{(d-1)})^\mathsf{T}\Lambda_d \\ v^\mathsf{T} & 0 & 0 & 0 & \Lambda_d W^{(d-1)} & -2\Lambda_d - S_d \end{bmatrix}.$$
$$(11)$$

With the above choice of $C$, we can immediately show the following EP-LipSDP for the multi-layer network case:

$$\min_{\zeta \geq 0, \lambda^{(k)} \geq 0, \tau^{(k)} \geq 0, \gamma \geq 0} \zeta + \left(2 + \sum_{k=1}^d \sum_{j=1}^{n_k} c_{kj}\right)\lambda_{\max}^+(C). \quad (12)$$

Now we can use a similar proof for Theorem 1 to show the following result for multi-layer networks.

**Theorem 2.** *Let $opt_4$ be the optimal value of Problem (10) and $opt_5$ be the optimal value of Problem (12). In addition, let $opt_6$ be the optimal value of the original LipSDP (9). We have $opt_4 = opt_5 = 2\sqrt{opt_6}$. In other words, all three programs give the same Lipschitz bounds.*

The proof is similar to the one of Theorem 1 presented in the appendix. Now we have obtained the exact penalty form of LipSDP which can be solved using first-order methods.

## 5   ADD-ON TECHNIQUES FOR IMPROVING PRACTICAL PERFORMANCES

In this section, we discuss a few techniques that can further improve the performance in terms of running time and practical convergence of first-order algorithms for solving EP-LipSDP in Section 4.

**Eigenvalue approximation.**   We formulate the Lipschitz constant estimation as an eigenvalue optimization problem that is differentiable almost everywhere. Consequently, estimating the extreme eigenvalue of a given matrix becomes an important sub-task. There exist various efficient algorithms to estimate the extreme value of a given matrix, and the efficiency can be significant improved via compromising the precision a little bit. In this work, we also consider the Lanczos algorithm (Lanczos, 1950) to compute the largest eigenvalue of a matrix as in Dathathri et al. (2020). Lanczos algorithm is an iterative algorithm used to find the extreme eigenvalues of symmetric matrices. It computes a small sub-matrix whose extreme eigenvalues are close to those of the original matrix. One can specify the size of the sub-matrix. The larger the submatrix is, the closer the eigenvalue is to the original value with a cost of higher computational resources.

**Sparse matrix multiplication.**   When using the Lanczos algorithm to compute the extreme eigenvalue of a matrix $A$, we only need access $x \to Ax$, instead of the explicit form of $A$. That is, we only need to know what is $Ax$ for any vector $x$. This provides further optimization opportunities when using the Lanczos algorithm. The matrix considered in this work has a fixed sparse pattern, i.e., it is almost a tridiagonal matrix, which encodes the computation of the (convolutional) neural network. For deep/wide (convolutional) networks, the SDP constraint matrix is very sparse. Thus, we can implement a native sparse matrix multiplication routine, which can significantly improve the efficiency of our method in addressing large networks on CIFAR10 and ImageNet.

**Analytical initialization.**   Theoretically, we can initialize the variables of our first-order iteration randomly and apply a (sub)gradient method for infinitely many steps with sufficiently small step size (learning rate) to ensure the convergence to the global minimum. In practice, we can only run the first-order subgradient method for finitely many steps. To ensure that our first-order method is better than the naive matrix norm product bound, we develop a special analytical initialization scheme. Specifically, we derive a specialized analytical solution to (11), which exactly recovers the naive norm product bound. This analytical solution is then used to initialize our subgradient method. Consequently, EP-LipSDP with such an initialization produces at least as tight Lipschitz bound as the matrix norm product. To derive this analytical initialization, we mainly use the Schur complement lemma and the block matrix inversion formula. The details are presented in Appendix B.

## 6   EVALUATION AND DISCUSSION

In this section, we conduct several experiments to empirically evaluate the algorithm proposed in this work. The evaluation aims to answer the following research questions: **(R1)** Does EP-LipSDP produce the same value as the original LipSDP when both problems are fully solved? **(R2)** Does our algorithm possess running time and memory advantage compared to LipSDP? **(R3)** How does each add-on technique from Section 5 contribute to the overall performance of EP-LipSDP?

Based on the PyTorch package, we developed a first-order implementation of EP-LipSDP, termed as **LipDiff**. The pseudocode is described in Appendix C. We use the ADAM optimizer (Kingma & Ba, 2015) as LipDiff's gradient descent optimizer and tune the number of gradient steps and step size (learning rate) accordingly for each problem. We also implemented the following variants (the details of these variants can be found in Appendix D):

**1. LipDiff-Ex** is the algorithm using exact eigenvalue instead of the Lanczos approximation.

Table 2: In this table, we present results comparing different variants of our approach to LipSDP. In MNIST-DNN experiment, if we compare the benchmark values from LipSDP and LipDiff-Ex, we can find that their values are very close. By tuning the parameters, we can get results fairly close to LipSDP value in a much shorter time. For MNIST-CNN, the SDP constraint size is of $4021 \times 4021$ and the Mosek solver used for LipSDP triggers the out-of-memory (OOM) error, while all other LipDiff variants can still run. Furthermore, LipDiff-Ex can reduce the Lipschitz constant estimation by 48% compared to the norm product method. For CIFAR10-CNN, LipDiff can reduce the Lipschitz constant by 58% compared to the norm product.

| Datasets | Models | | Product | LipSDP | LipDiff | LipDiff-Ex | LipDiff-Dense | LipDiff-Rand |
|---|---|---|---|---|---|---|---|---|
| **MNIST** | **DNN** | Result | 9.31 | 4.82 | 4.90 | 4.86 | 4.96 | 5.89 |
| | | Time (s) | 0.13 | 54.57 | 28.69 | 19.27 | 12.48 | 29.27 |
| | | Memory (MB) | 1.54 | 169.83 | 118 | 114 | 114 | 118 |
| **MNIST** | **CNN** | Result | 24.79 | OOM | 14.76 | 13.08 | 14.66 | 2201.66 |
| | | Time (s) | 0.16 | – | 178.99 | 559.08 | 185.34 | 186.89 |
| | | Memory (MB) | 18.77 | – | 2640 | 1534 | 1536 | 2640 |
| **CIFAR10** | **CNN** | Result | 35.45 | OOM | 14.82 | 18.52 | 16.12 | 1731.82 |
| | | Time (s) | 98.08 | – | 2777.07 | 36000 | 25126.01 | 2723.02 |
| | | Memory (GB) | 0.51 | – | 60.05 | 51.39 | 51.41 | 60.05 |

2. **LipDiff-Dense** implements the Lanczos algorithm with explicit matrix multiplication.
3. **LipDiff-Rand** initializes the algorithm from a random initialization rather than the analytical solution and also implements the Lanczos algorithm in the sparse matrix multiplication way.

**Baseline.** We use two main baselines: 1. **Product**, the weight matrix norm product, which provides a naive upper bound for the Lipschitz constant of the neural network, and is currently the only method for large networks on ImageNet (Leino et al., 2021; Hu et al., 2023); 2. **LipSDP** (Fazlyab et al., 2019), which is implemented using the CVXPY package (Diamond & Boyd, 2016) and relies on the Mosek solver (ApS, 2019) to solve the SDP. In our experiments, the LipSDP implementation is run on CPU. In contrast, the LipDiff variants are GPU-friendly, and hence run on GPU.

**First Part of Experiments: MNIST and CIFAR10..** To provide good sanity checks, we first run our algorithms on three networks: 1. an MNIST (LeCun & Cortes, 2010) dense network, with a single hidden layer of 128 ReLU nodes; 2. An MNIST convolutional network, with 1 convolutional layer, and 2 fully connected layers; 3. CIFAR10 convolutional network with 3 convolutional layers and 3 fully connected layers. These networks are summarized in Table 1.

Each of the networks has 10 classes. We will compute the Lipschitz constant of the function corresponding to 8th label as in the evaluation of Wang et al. (2022). For the evaluation, we set the timeout for each experiment to be 10 hours. If the running time is over 10 hours, we will

Table 1: Models used for the experiments.

| Model | Structure | Parameters | Acc. | SDP size |
|---|---|---|---|---|
| MNIST-DNN | 1FC | 203 530 | 97% | $1041 \times 1041$ |
| MNIST-CNN | 1C2FC | 314 982 | 98% | $4021 \times 4021$ |
| CIFAR10-CNN | 3C3FC | 1 344 298 | 80% | $22 529 \times 22 529$ |

terminate the program and report the best number. To answer RQ1 and RQ2, we will compare the performances of LipSDP and LipDiff variants in terms of the results, and memory and running time needed for the computation. To answer RQ3, we will compare the results of LipDiff variants. The results of the evaluation for this part are summarized in Table 2.

**RQ1.** If we compare the result for MNIST-DNN in table 2, we can find that LipDiff-Ex computes almost the same value as LipSDP does. In addition, LipDiff and LipDiff-Dense produce similar results as LipSDP's value. These results provide an affirmative answer to RQ1, that our eigenvalue optimization formulation gives the same value as LipSDP, as showed in Theorem 1.

**RQ2.** In terms of *memory consumption*, because LipDiff variants are run on GPU while LipSDP is run on CPU with different package implementations, a direct comparison between the numbers is not sensible. However, we can still compare between experiments. Recall that the SDP constraints for MNIST-DNN, MNIST-CNN and CIFAR10-CNN are of sizes $1041 \times 1041$, $4021 \times 4021$ and $22529 \times 22529$. As we can observe from Table 2, the memory consumption for LipDiff roughly scales linearly to the size of the SDP constraint. However, for LipSDP, the memory requirement is much more beyond linear. For example, the total memory of the server is 528 GB, 4000 times larger than 118 MB, and LipSDP cannot even work on a network with the SDP constraint size $4021 \times 4021$,

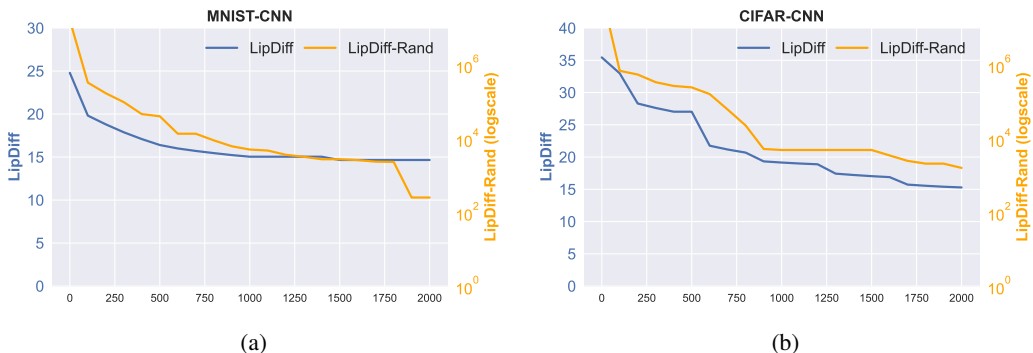

(a)          (b)

Figure 1: These plots show the minimization of LipDiff and LipDiff-Rand on MNIST-CNN Figure 1a and CIFAR-CNN models Figure 1b. The left axis of each plot shows LipDiff (**blue**) and the right axis shows LipDiff-Rand (**orange**) in log scale. The x-axis describes the number step of the optimization. We observe that LipDiff obtains a much lower value than LipDiff-Rand due to its better initialization. Additionally, LipDiff shows a sharp decrease at the beginning of the optimization, making it an efficient method for improving over the product bound when the time budget is limited.

which is roughly 16 times larger than the MNIST-DNN. This comparison establishes the memory efficiency of LipDiff. For *running time*, because we can tune the number of steps and step size for gradient descent, we can obtain a fairly decent result within a short time. This tuning process offers more flexibility than a closed-form solver.

**RQ3.** From all experiments in Table 2, we can draw a conclusion that with the analytical solution initialization, LipDiff is consistently better than LipDiff-Random. This is particularly true when the network is large and it is hard for the subgradient method from a bad initial state to converge with many free variables. For approximating extreme eigenvalue using the Lanczos algorithm, we always save significant running time, at a price of a little potential precision loss. When using the sparse matrix representation for matrix multiplication, the benefit is only significant when the network is large enough. For small networks such as the MNIST DNN, matrix multiplication using dense representation is faster than the sparse representation. The reason is that for small networks, the SDP constraint is almost always dense rather than sparse. However, at the scale of the CIFAR10 network, the sparse matrix multiplication can save almost 90% of running time. For memory consumption, we can consistently find that LipDiff-Ex uses least memory and LipDiff uses the most. This comes from our design choice: For the implementations using Lanczos algorithm to approximate the maximum eigenvalue, we also include the original constraint to compute the exact eigenvalue. Therefore, the Lanczos tools would compute an extra Lanczos submatrix. The sparse representation will further implement an additional sparse multiplication routine and thus consume even more memory.

**Second Part of Experiments: ImageNet .** To fully test the scalability of our proposed method, we additionally applied LipDiff on the pretrained AlexNet (Krizhevsky et al., 2017), which was designed for ImageNet classification (Deng et al., 2009). AlexNet is composed of a feature extraction sub-network and a classification sub-network. Noticeably, AlexNet is much larger than all the MNIST/CIFAR10 networks considered in our previous experiments. To address a network at this scale, we need to combine all the add-on techniques in Section 5. Importantly, we use the sub-multiplicity of norms to decompose AlexNet, and apply LipDiff on the sub-networks, which eventually reduces the total Lipschitz constant of AlexNet by 73%, compared to the matrix norm product bound. To the best of our knowledge, this is the first time that LipSDP is successfully scaled to ImageNet. More details of our ImageNet experiments can be found in Appendix H.

**Conclusion.** Overall, we can conclude that LipDiff is much more memory-efficient than the original default interior point implementation of LipSDP. However, without our specialized analytical solution, it is hard for LipDiff to converge from a random initialization. With all the add-on techniques proposed in our work, for the first time, we can estimate the Lipschitz constant of large networks on CIFAR10 and ImageNet, and this estimation is always at least as good as the naive matrix norm bound. Our algorithm can terminate at any time, which further offers more flexibility for users with different time budgets.

ACKNOWLEDGEMENT

Z. Wang and S. Jha are partially supported by DARPA under agreement number 885000, NSF CCF-FMiTF-1836978 and ONR N00014-21-1-2492. A. Havens and B. Hu are generously supported by the NSF award CAREER-2048168 and the AFOSR award FA9550-23-1-0732. Y. Zheng is partially supported by NSF ECCS-2154650 and NSF CMMI-2320697. Y. Chen is partially supported by NSF CCF-1704828 and NSF CCF-2233152.

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

## A TECHNICAL PROOFS

In this section, we provide a detailed proof for Theorem 1.

*Proof.* We first prove $\mathbf{opt}_2 \leq \mathbf{opt}_1$. This direction is straightforward. Suppose that an optimal solution to Problem (6) is $(\zeta^*, \lambda^*, \tau^*, \gamma^*)$. By definition, this solution is also feasible to Problem (7). Furthermore, the corresponding matrix $C \preceq 0$, and thus $\lambda_{\max}^+(C) = 0$. Hence we have $\mathbf{opt}_2 \leq \mathbf{opt}_1$.

Next, we prove $\mathbf{opt}_1 \leq \mathbf{opt}_2$. Suppose an optimal solution to Problem (7) is $(\zeta^0, \tau^0, \lambda^0, \gamma^0)$. If this solution makes the corresponding matrix $C \preceq 0$, then it is also feasible to Problem (6) with the same cost value $\frac{1}{2}\zeta^0$. In this case, we have $\mathbf{opt}_1 \leq \mathbf{opt}_2$. If $\lambda_{\max}(C) = \alpha > 0$, then this solution is not feasible to Problem (6). However, we can construct another optimal solution by exploiting the structure of $C$. Specifically, we choose

$$\gamma' = \gamma^0 + \alpha, \quad \lambda_i' = \lambda_i^0 + \alpha \quad \zeta' = \zeta^0 + \big(2 + \sum_{j=1}^m a_j\big)\alpha, \quad \tau' = \tau^0. \tag{13}$$

We next show that this new solution is optimal for Problem (7) and also feasible for Problem (6). Denote $\Lambda_1' = \operatorname{diag}(\tau')$, $\Lambda_2' = \operatorname{diag}(\lambda')$, $\Lambda_1^0 = \operatorname{diag}(\tau^0)$, and $\Lambda_2^0 = \operatorname{diag}(\lambda^0)$. Recall the structure of $C$ and we have

$$\begin{bmatrix} \sum_{j=1}^m a_j \lambda_j' + \gamma' - \zeta' & 0 & v \\ 0 & -\gamma' I & W^\mathsf{T} \Lambda_1' \\ v^\mathsf{T} & \Lambda_1' W & -2\Lambda_1' - \Lambda_2' \end{bmatrix}$$
$$= \begin{bmatrix} \sum_{j=1}^m a_j \lambda_j^0 + \gamma^0 - \zeta^0 & 0 & v \\ 0 & -\gamma^0 I & W^\mathsf{T} \Lambda_1^0 \\ v^\mathsf{T} & \Lambda_1^0 W & -2\Lambda_1^0 - \Lambda_2^0 \end{bmatrix} - \alpha I \preceq 0.$$

Thus, the new solution in (13) is feasible for Problem (6). Furthermore, it is easy to verify that the new solution in (13) has the same cost as the solution $(\zeta^0, \tau^0, \lambda^0, \gamma^0)$ in Problem (7), which is

$$\frac{\zeta'}{2} = \frac{\zeta^0}{2} + \frac{1}{2}\big(2 + \sum_{j=1}^m a_j\big)\alpha.$$

We thus also have $\mathbf{opt}_1 \leq \mathbf{opt}_2$.

The proof of $\mathbf{opt}_1 = 2\sqrt{\mathbf{opt}_3}$ can be seen from the equivalence of various forms of LipSDP, i.e. SDP forms (1), (3), and (6). See a detailed proof for the equivalence of (1), (3), and (6) in Appendix E. Now our proof is complete. □

Upon choosing $\rho > 2 + \sum_{j=1}^n a_j$ as the penalty parameter in Problem (7), one can further show that any optimal solution to (7) makes the corresponding $C \preceq 0$, thus the second case in the proof above never happens. We put this result as a lemma below.

**Lemma 1.** *Let $\rho > 2 + \sum_{j=1}^n a_j$ as the penalty parameter in Problem (7). Then any optimal solution to (7) makes the corresponding $C \preceq 0$.*

*Proof.* Let $(\zeta^0, \lambda^0, \tau^0, \gamma^0)$ be an arbitrary optimal solution to Problem (7). For the sake of deriving contradiction, suppose the corresponding $C^0$ has $\lambda_{\max}(C^0) = \alpha > 0$. We construct another solution $(\zeta', \tau', \lambda', \gamma')$ by

$$\gamma' = \gamma^0 + \alpha, \quad \lambda_i' = \lambda_i^0 + \alpha, \quad \zeta' = \zeta^0 + \Big(\sum_{j=1}^m a_j + 2\Big)\alpha, \quad \tau' = \tau^0. \tag{14}$$

Then the corresponding $C'$ satisfies

$$C' = \begin{bmatrix} \sum_{j=1}^m a_j \lambda_j' + \gamma' - \zeta' & 0 & v \\ 0 & -\gamma' I & W^\mathsf{T} \Lambda_1' \\ v^\mathsf{T} & \Lambda_1' W & -2\Lambda_1' - \Lambda_2' \end{bmatrix}$$
$$= \begin{bmatrix} \sum_{j=1}^m a_j \lambda_j^0 + \gamma^0 - \zeta^0 & 0 & v \\ 0 & -\gamma^0 I & W^\mathsf{T} \Lambda_1^0 \\ v^\mathsf{T} & \Lambda_1^0 W & -2\Lambda_1^0 - \Lambda_2^0 \end{bmatrix} - \alpha I \preceq 0.$$

When $\rho > 2 + \sum_{j=1}^{n} a_j$, we find that $(\zeta', \tau', \lambda', \gamma')$ has the strictly lower cost than $(\zeta^0, \tau^0, \lambda^0, \gamma^0)$:

$$\frac{\zeta'}{2} + \rho\lambda_{\max}^{+}(C') = \frac{\zeta^0}{2} + \frac{1}{2}\left(2 + \sum_{j=1}^{m} a_j\right)\alpha + 0 < \frac{\zeta^0}{2} + \rho\alpha = \frac{\zeta^0}{2} + \rho\lambda_{\max}^{+}\left(C^0\right),$$

which contradicts the optimality of $(\zeta^0, \tau^0, \lambda^0, \gamma^0)$ to Problem (7). □

## B ANALYTICAL SOLUTION FOR THE EIGENVALUE PROBLEM

Let us start with the two-layer network and derive an initial solution to the SDP problem (Equation (7)). We use the SDP for a two-layer network as an example. Recall that

$$C = \begin{bmatrix} \sum_{j=1}^{m} a_j\lambda_j + \gamma - \zeta & 0 & v \\ 0 & -\gamma I & W^{\mathsf{T}}\Lambda_1 \\ v^{\mathsf{T}} & \Lambda_1 W & -2\Lambda_1 - \Lambda_2 \end{bmatrix},$$

where $\Lambda_1 = \mathrm{diag}(\tau)$, and $\Lambda_2 = \mathrm{diag}(\lambda)$. Clearly, $\Lambda_2$ comes from redundant constraints. Now we can let $\lambda = 0$, then we basically reduce $C$ to a similar constraint of LipSDP, then we want to find an analytical solution to

$$C = \begin{bmatrix} \gamma - \zeta & 0 & v \\ 0 & -\gamma I & W^{\mathsf{T}}\Lambda_1 \\ v^{\mathsf{T}} & \Lambda_1 W & -2\Lambda_1 \end{bmatrix} \preceq 0,$$

and minimize $\zeta$.

Let $A = \begin{bmatrix} \gamma I & -W^{\mathsf{T}}\Lambda_1 \\ -\Lambda_1 W & 2\Lambda_1 \end{bmatrix}$.

By the Schur complement lemma, to have $C \preceq 0$, we need $A \succ 0$ and

$$\zeta - \gamma - (0, -v)A^{-1}((0, -v))^{\mathsf{T}} \geq 0.$$

We use $(A^{-1})_{22}$ to denote the second row and second column block matrix of $A^{-1}$.

Again, by the Schur complement lemma, to have $A \succ 0$, we need $\gamma > 0$ and

$$2\Lambda_1 - \Lambda_1 W^{\mathsf{T}}W\Lambda_1/\gamma \succeq 0.$$

We can set $\Lambda_1 = aI$, and $\gamma = b\|W\|_{\mathrm{op}}^2$:

$$2aI - \frac{a^2}{b\|W\|_{\mathrm{op}}^2}W^{\mathsf{T}}W \succeq a(2 - \frac{a}{b})I.$$

Then we need $2b > a$.

From the block matrix inversion formula, we have

$$(A^{-1})_{22} = (2T - TW^{\mathsf{T}}WT/\gamma)^{-1}.$$

With the choice of $\Lambda_1$ and $\gamma$, we have

$$\left\|(A^{-1})_{22}\right\|_{\mathrm{op}} \leq \left(a(2 - \frac{a}{b})\right)^{-1} = \frac{b}{a(2b - a)}.$$

Then

$$v(A^{-1})_{22}v^{\mathsf{T}} \leq \frac{b}{a(2b - a)}\|v\|.$$

To have $\zeta - \gamma - v(A^{-1})_{22}v^{\mathsf{T}} \geq 0$, we need

$$\zeta - \gamma - \frac{b}{a(2b - a)}\|v\|^2 \geq 0,$$

then we need

$$\zeta = b \left\| W \right\|_{\mathrm{op}}^2 + \frac{b}{a(2b-a)} \left\| v \right\|^2.$$

We can let $a = b = \frac{\|v\|}{\|W\|_{\mathrm{op}}}$, then we have

$$\zeta = 2 \left\| W \right\|_{\mathrm{op}} \left\| v \right\|.$$

Because we will divide the $\zeta$ by 2, this recovers the naive upper bound of $\ell_2$ Lipshitz constant: $\left\| W \right\|_{\mathrm{op}} \left\| v \right\|$.

For general multi-layer networks, the derivation is essentially the same as the two-layer network case. It is a more careful recursive application of the Schur complement lemma and the block matrix inversion formula. Let us use a three-layer network to illustrate the recursive structure for multi-layer networks.

We have

$$C = \begin{bmatrix} \gamma - \zeta & 0 & 0 & v \\ 0 & -\gamma I & W_1^\mathsf{T} \Lambda_1 & 0 \\ 0 & \Lambda_1 W_1 & -2\Lambda_1 & W_2 \Lambda_2 \\ v^\mathsf{T} & 0 & \Lambda_2 W_2 & -2\Lambda_2 \end{bmatrix} \preceq 0$$

and minimize $\zeta$.

Let

$$C_1 = \begin{bmatrix} \gamma I & -W_1^\mathsf{T} \Lambda_1 & 0 \\ -\Lambda_1 W_1 & 2\Lambda_1 & -W_2 \Lambda_2 \\ 0 & -\Lambda_2 W_2 & 2\Lambda_2 \end{bmatrix}$$

and

$$C_2 = \begin{bmatrix} \gamma I & -W_1^\mathsf{T} \Lambda_1 \\ -\Lambda_1 W_1 & 2\Lambda_1 \end{bmatrix}.$$

Let

$$A_1 = \begin{bmatrix} \gamma I & -W_1^\mathsf{T} \Lambda_1 & 0 \\ -\Lambda_1 W_1 & \Lambda_1 & 0 \\ 0 & 0 & 0 \end{bmatrix}$$

and

$$A_2 = \begin{bmatrix} 0 & 0 & 0 \\ 0 & \Lambda_1 & -W_2 \Lambda_2 \\ 0 & -\Lambda_2 W_2 & 2\Lambda_2 \end{bmatrix},$$

so $A_1 + A_2 = C_1$.

From the Schur complement lemma, we need

$$\zeta - \gamma - (0, 0, -v) C_1^{-1} (0, 0, v)^\mathsf{T} \geq 0$$

and $C_1 \succ 0$.

To have $C_1 \succ 0$, we can have $A_1 \succeq 0$ and $A_2 \succeq 0$ (we need at least one strict inequality, and our final assignment of the values will achieve this). To have $A_1 \succeq 0$, we have $\gamma > 0$ and

$$\Lambda_1 - \Lambda_1 W_1^\mathsf{T} W_1 \Lambda_1 / \gamma \succeq 0.$$

We can set $\gamma = a(\left\| W_1 \right\|_{\mathrm{op}} \left\| W_2 \right\|_{\mathrm{op}})^2$ and $\Lambda_1 = b \left\| W_2 \right\|_{\mathrm{op}}^2 I$:

$$b \left\| W_2 \right\|_{\mathrm{op}}^2 I - \frac{b^2 \left\| W_2 \right\|_{\mathrm{op}}^2}{a \left\| W_1 \right\|_{\mathrm{op}}^2} W_1^\mathsf{T} W_1 \succeq b \left\| W_2 \right\|_{\mathrm{op}}^2 (1 - \frac{b}{a}) I.$$

To have $A_2 \geq 0$, we need $b > 0$ and

$$2\Lambda_2 - \Lambda_2 W_2^\mathsf{T} W_2 \Lambda_2 / (b \left\| W_2 \right\|_{\mathrm{op}}^2) \succeq 0.$$

We can set $\Lambda_2 = cI$:

$$2cI - \frac{c^2}{b\|W_2\|_{op}^2}W_2^\mathsf{T}W_2 \succeq c(2 - \frac{c}{b})I.$$

As a result, we only need $b \leq a$ and $c \leq 2b$.

To have

$$\zeta - \gamma - (0,0,-v)C_1^{-1}(0,0,v)^\mathsf{T} \geq 0,$$

we need to use the block inversion formula:

$$[C_1^{-1}]_{33} = (-2\Lambda_2 - (0,-\Lambda_2W_2)C_2^{-1}(0,-\Lambda_2W_2)^\mathsf{T})^{-1}.$$

$$[C_2^{-1}]_{22} = (2\Lambda_1 - W_1^\mathsf{T}\Lambda_1\Lambda_1W_1/\gamma)^{-1}.$$

$$[C_1^{-1}]_{33} = (2cI - (\Lambda_2W_2(2\Lambda_1 - W^\mathsf{T}\Lambda_1\Lambda_1W_1/\gamma)^{-1}W_2^\mathsf{T}\Lambda_2))^{-1}.$$

$\left\|[C_1^{-1}]_{33}\right\|_{op} = \left\|(2cI - (\Lambda_2W_2(2\Lambda_1 - W_1^\mathsf{T}\Lambda_1\Lambda_1W_1/\gamma)^{-1}W_2^\mathsf{T}\Lambda_2))^{-1}\right\|_{op} \leq (2c - c^2\|W_2\|_{op}^2(2b\|W_2\|_{op}^2 - b^2\|W_2\|_{op}^2/a)^{-1})^{-1}.$

Notice that this has recursive inequalities to make the constraint work: $\left\|WTTW^\mathsf{T}\right\|_{op} \leq c\|W\|_{op}$.

To have

$$\zeta - \gamma - v(C_1^{-1})_{33}v^\mathsf{T} \geq 0,$$

we only need

$$\zeta = a(\|W_1\|_{op}\|W_2\|_{op})^2 + \|v\|^2(2c - c^2\|W_2\|_{op}^2(2b\|W_2\|_{op}^2 - b^2\|W_2\|_{op}^2/a)^{-1})^{-1}.$$

We set $a = b = c = \frac{\|v\|}{\|W_1\|_{op}\|W_2\|_{op}}$. This would recover the naive bound of $\|W_1\|_{op}\|W_2\|_{op}\|v\|$.

More generally, if we have a general multiple-layer network, we only need to apply the Schur complement lemma and the matrix inverse formula recursively:

$$\begin{bmatrix} \sum(a_j^{(i)})\lambda_j + \gamma - \zeta & 0 & 0 & \ldots & v \\ 0 & -\gamma I & W^{(1)}\Lambda_1 & \ldots & 0 \\ \vdots & \ldots & \ldots & \ddots & \vdots \\ v^\mathsf{T} & 0 & \ldots & \Lambda_{d-1}(W^{d-1})^\mathsf{T} & -2\Lambda_{d-1} - S_{d-1} \end{bmatrix}.$$

We can set all $S = 0$, and for each $\Lambda_j = \Pi_{i=j+1}^{d-1}c\|W_i\|_{op}^2$, where $c = \frac{\|v\|}{\Pi_{i=1}^{d-1}c\|W_i\|_{op}}$. $\gamma = \|v\|\Pi_{i=j+1}^{d-1}c\|W_i\|_{op}$, and $\zeta = 2\|v\|\Pi_{i=1}^{d-1}c\|W_i\|_{op}$.

## C  PSEUDOCODE OF LIPDIFF

Here we provide the pseudo-code of the LipDiff algorithm (Algorithm 1). Notice that one can substitute the direct ADAM optimizer with more complicated scheduling methods on learning rates and other optimizers such as the stochastic gradient descent optimizer.

## D  OTHER EXPERIMENTAL SPECIFICATIONS

**LipDiff variants specification.** **LipDiff** uses all the optimization techniques proposed in Section 5. **LipDiff-Ex** uses the explicit matrix formulation to exactly compute the eigenvalue, thus it is a dense representation, and also it applies the analytical initialization. **LipDiff-Dense** uses eigenvalue approximation but implements it using dense matrix multiplication. **LipDiff-Rand** uses the sparse

---

**Algorithm 1** The LipDiff Algorithm

---

     **Input**: The weight matrices $[W_i]_{i=1}^d$ of a neural network.
     **Output**: The Lipschitz constant of the neural network.
     **Hyperparameters**: Number of Iterations $n$; Lanczos steps $l$; Step size $\alpha$

 1: Declare decision/dual variables as in Equation (11) and assign their values according to Appendix B
 2: Create an ADAM optimizer for the dual variables with learning rate $\alpha$
 3: **for** $n$ iterations **do**
 4:     Create the SDP constraint Equation (11) with $[W_i]_{i=1}^d$ and the dual variables.
 5:     Compute the Lanczos submatrix $L$ of size $l \times l$ of $C$
 6:     Define the loss function as Equation (12)
 7:     Compute the subgradient of the loss function via autodiff and update the dual variables
 8: **end for**
 9: Return the loss value

---

matrix multiplication to approximate the eigenvalue but initializes the algorithm with random assignments. We made the following design choice: For the implementations using the Lanczos algorithm to approximate the maximum eigenvalue, we also include the original constraint to compute the exact eigenvalue, and all the reported numbers come from the exact eigenvalue.

**Server specification.** All the experiments are run on a server with thirty-two AMD EPYC 7313P 16-core processors, 528 GB of memory, and four Nvidia A100 GPUs. Each GPU has 80 GB of memory.

**Training specification.** We train all the networks with the SGD optimizer with a learning rate 0.02, momentum 0.9, and weight decay of 0.0008. We train MNIST networks for 50 epochs, and CIFAR10 networks for 200 epochs.

**LipDiff specifications.** For LipDiff, we apply the interval scheduling of step sizes. We divide the number of iterations into $n$ groups. Within each group, we start from a step size $\alpha$ and exponentially decay the step size to $\alpha/10$.

For MNIST-DNN, we run LipDiff for a total of 1000 iterations, and divide the 1000 iterations into 2 groups. The step size is initialized to 0.04. We use Lanczos steps 15.

For MNIST-CNN, we run LipDiff for a total of 2000 iterations, and divide the 1000 iterations into 4 groups. The step size is initialized to 0.05. We use Lanczos steps 30.

For CIFAR10-CNN, we run LipDiff for a total of 2000 iterations, and divide the 2000 iterations to 5 groups. The step size is initialized to 0.06. We use Lanczos steps 50.

# E   On the equivalence of different forms of LipSDP

## E.1   On the Equivalence of SDP Forms (1) and (3)

In this section, we establish the equivalence between (1) and (3). To make our proof more readable, we slightly change the notation of (1). Specifically, (1) is exactly the same as the following problem:

$$\min_{\hat{\zeta}, \hat{\tau}} \quad \hat{\zeta}$$

$$\text{subject to} \quad \begin{bmatrix} -\hat{\zeta}I & W^\mathsf{T}\operatorname{diag}(\hat{\tau}) \\ \operatorname{diag}(\hat{\tau})W & -2\operatorname{diag}(\hat{\tau}) + v^\mathsf{T}v \end{bmatrix} \preceq 0, \quad \hat{\tau} \geq 0, \tag{15}$$

We restate (3) below

$$
\min_{\zeta,\tau,\gamma} \quad \zeta
$$
$$
\text{subject to} \quad
\begin{bmatrix}
\gamma - \zeta & 0 & v \\
0 & -\gamma I & W^\mathsf{T} \operatorname{diag}(\tau) \\
v^\mathsf{T} & \operatorname{diag}(\tau)W & -2\operatorname{diag}(\tau)
\end{bmatrix}
\preceq 0, \tag{16}
$$
$$
\zeta \geq 0,\ \tau \geq 0,\ \gamma \geq 0.
$$

Next, we will prove the following result which establishes the equivalence of (15) and (16).

**Proposition 1.** *Denote the optimal values of (15) and (16) as $\hat{\zeta}^*$ and $\zeta^*$, respectively. Then $\zeta^* = 2\sqrt{\hat{\zeta}^*}$.*

*Proof.* If $v = 0$, then we trivially have $\zeta^* = 2\sqrt{\hat{\zeta}^*} = 0$. For the rest of the proof, we assume that $v$ is not a zero vector. In this case, any feasible solution to (16) implies that $\gamma - \zeta < 0$.

By Schur complement, (16) is equivalent to the following problem:

$$
\min_{\zeta,\tau,\gamma} \quad \zeta
$$
$$
\text{subject to} \quad
\begin{bmatrix}
-\gamma I & W^\mathsf{T} \operatorname{diag}(\tau) \\
\operatorname{diag}(\tau)W & -2\operatorname{diag}(\tau) + \frac{1}{\zeta-\gamma}v^\mathsf{T}v
\end{bmatrix}
\preceq 0, \tag{17}
$$
$$
\gamma < \zeta,\ \zeta \geq 0,\ \tau \geq 0,\ \gamma \geq 0.
$$

In other words, the optimal solution for (17) is also given by $\zeta^*$.

Let $(\hat{\zeta}^*, \hat{\tau}^*)$ be the optimal feasible point for (15). If we choose $\gamma = \sqrt{\hat{\zeta}^*}$, $\zeta = 2\gamma = 2\sqrt{\hat{\zeta}^*}$, and $\tau = \gamma\hat{\tau}^* = \sqrt{\hat{\zeta}^*}\hat{\tau}^*$, then we have

$$
\begin{bmatrix}
-\gamma I & W^\mathsf{T} \operatorname{diag}(\tau) \\
\operatorname{diag}(\tau)W & -2\operatorname{diag}(\tau) + \frac{1}{\zeta-\gamma}v^\mathsf{T}v
\end{bmatrix}
= \frac{1}{\sqrt{\hat{\zeta}^*}}
\begin{bmatrix}
-\hat{\zeta}^* I & W^\mathsf{T} \operatorname{diag}(\hat{\tau}^*) \\
\operatorname{diag}(\hat{\tau}^*)W & -2\operatorname{diag}(\hat{\tau}^*) + v^\mathsf{T}v
\end{bmatrix}
\preceq 0.
$$

Therefore, the above choice of $(\gamma, \zeta, \tau)$ must be a feasible point for (17). Consequently, the optimal value of (17) must be upper bounded by the value of $\zeta$ in this feasible point, which is $\zeta = 2\gamma = 2\sqrt{\hat{\zeta}^*}$. Therefore, we have $\zeta^* \leq 2\sqrt{\hat{\zeta}^*}$

Next, we show $\zeta^* \geq 2\sqrt{\hat{\zeta}^*}$. Let us introduce the following problem

$$
\min_{\zeta,\tau,\gamma} \quad 2\sqrt{\gamma(\zeta - \gamma)}
$$
$$
\text{subject to} \quad
\begin{bmatrix}
-\gamma I & W^\mathsf{T} \operatorname{diag}(\tau) \\
\operatorname{diag}(\tau)W & -2\operatorname{diag}(\tau) + \frac{1}{\zeta-\gamma}v^\mathsf{T}v
\end{bmatrix}
\preceq 0, \tag{18}
$$
$$
\gamma < \zeta,\ \zeta \geq 0,\ \tau \geq 0,\ \gamma \geq 0.
$$

This problem has the same feasible set as (17). Note that, for any point in the feasible set, we must have $2\sqrt{\gamma(\zeta - \gamma)} \leq \zeta$ (this is equivalent to $\zeta^2 - 4\zeta\gamma + 4\gamma^2 \geq 0$). Therefore, considering the same feasible point, the objective function of (18) is always upper bounded by the objective function of (17). Denote the optimal solutions of (18) and (17) as $(\zeta^\dagger, \tau^\dagger, \gamma^\dagger)$ and $(\zeta^*, \tau^*, \gamma^*)$, respectively. Clearly, $(\zeta^*, \tau^*, \gamma^*)$ is also a feasible point for (18). Then we must have

$$
2\sqrt{\gamma^\dagger(\zeta^\dagger - \gamma^\dagger)} \leq 2\sqrt{\gamma^*(\zeta^* - \gamma^*)} \leq \zeta^*. \tag{19}
$$

Finally, if we choose $\hat{\tau} = (\zeta^\dagger - \gamma^\dagger)\tau^\dagger$ and $\hat{\zeta} = \gamma^\dagger(\zeta^\dagger - \gamma^\dagger)$. Then we have

$$
\begin{bmatrix}
-\hat{\zeta} I & W^\mathsf{T} \operatorname{diag}(\hat{\tau}) \\
\operatorname{diag}(\hat{\tau})W & -2\operatorname{diag}(\hat{\tau}) + v^\mathsf{T}v
\end{bmatrix}
= (\zeta^\dagger - \gamma^\dagger)
\begin{bmatrix}
-\gamma^\dagger I & W^\mathsf{T} \operatorname{diag}(\tau^\dagger) \\
\operatorname{diag}(\tau^\dagger)W & -2\operatorname{diag}(\tau^\dagger) + \frac{1}{\zeta^\dagger-\gamma^\dagger}v^\mathsf{T}v
\end{bmatrix}
\preceq 0
$$

Therefore, this choice of $(\hat{\zeta}, \hat{\tau})$ gives a feasible point for (15), and we have

$$\hat{\zeta}^* \le \hat{\zeta} = \gamma^\dagger(\zeta^\dagger - \gamma^\dagger). \tag{20}$$

A trivial combination of (19) and (20) leads to the desired conclusion $\zeta^* \ge 2\sqrt{\hat{\zeta}^*}$.

To summarize, we have established that $\zeta^* = 2\sqrt{\hat{\zeta}^*}$. Our proof is now complete. $\qquad\square$

**Remark 1.** *The above proof gives the right scaling between $\zeta^*$ and $\hat{\zeta}^*$. Now we provide a more intuitive treatment to explain how the factor of 2 appears in the scaling. Specifically, as mentioned in our main paper, if the semidefinite constraint in (3) is feasible, then the neural network is $\frac{\xi}{2}$-Lipschitz. Now we provide a proof for this fact. Recall that we have $f(x) = v\sigma(Wx + b^0) + b^1$, where $f(x)$ is a scalar. Obviously, the tightest Lipschitz bound is given by*

$$L_{\min} = \max_{x',x} \frac{|f(x') - f(x)|}{\|x' - x\|}. \tag{21}$$

*For any $L \ge L_{\min}$, we have $|f(x') - f(x)| \le L\|x' - x\| \ \forall x', x$. By using the slope-restricted property of $\sigma$, one upper bound for $L_{\min}$ is provided by the solution of the following problem which basically replaces $\sigma$ with the quadratic constraint in Fazlyab et al. (2019, section 2.2):*

$$\max_{\Delta x, \Delta z} \frac{|v\Delta z|}{\|\Delta x\|} \quad s.t. \begin{bmatrix} W\Delta x \\ \Delta z \end{bmatrix}^\mathsf{T} \begin{bmatrix} 0 & \operatorname{diag}(\tau) \\ \operatorname{diag}(\tau) & -2\operatorname{diag}(\tau) \end{bmatrix} \begin{bmatrix} W\Delta x \\ \Delta z \end{bmatrix} \ge 0, \tag{22}$$

*where $\Delta z = \sigma(Wx' + b^0) - \sigma(Wx + b^0)$, $\Delta x = x' - x$, and $\tau$ is any vector whose entrieis are all non-negative. If we scale $\Delta x$ and $\Delta v$ with a common factor, the constraint in (22) is maintained, and the cost remains unchanged . Therefore, (22) is equivalent to the following problem*

$$\max_{\Delta x, \Delta z} v\Delta z \quad s.t. \begin{bmatrix} W\Delta x \\ \Delta z \end{bmatrix}^\mathsf{T} \begin{bmatrix} 0 & \operatorname{diag}(\tau) \\ \operatorname{diag}(\tau) & -2\operatorname{diag}(\tau) \end{bmatrix} \begin{bmatrix} W\Delta x \\ \Delta z \end{bmatrix} \ge 0, \ \|\Delta x\| = 1.$$

*Notice that the absolute value in the objective is removed due to the fact that scaling $(\Delta x, \Delta z)$ with $-1$ does not affect feasibility. If we replace the equality constraint $\|\Delta x\| = 1$ with an inequality constraint $\|\Delta x\| \le 1$, then we get the following problem whose solution is an upper bound for (22).*

$$\max_{\Delta x, \Delta z} v\Delta z \quad s.t. \begin{bmatrix} W\Delta x \\ \Delta z \end{bmatrix}^\mathsf{T} \begin{bmatrix} 0 & \operatorname{diag}(\tau) \\ \operatorname{diag}(\tau) & -2\operatorname{diag}(\tau) \end{bmatrix} \begin{bmatrix} W\Delta x \\ \Delta z \end{bmatrix} \ge 0, \ \|\Delta x\| \le 1. \tag{23}$$

*Denote the optimal value of the above problem as $L^u$. Then $L^u$ is an upper bound for $L_{\min}$. It is obvious that any upper bound for $L^u$ will also be $L_{\min}$. Next, we will show that if the semidefinite constraint in (3) is feasible, then we have $\frac{\zeta}{2} \ge L^u \ge L_{\min}$, and hence $f$ is $\frac{\zeta}{2}$-Lipschitz. Specifically, if the semidefinite constraint in (3) is feasible, we immediately have*

$$\begin{bmatrix} 1 \\ \Delta x \\ \Delta z \end{bmatrix}^\mathsf{T} \begin{bmatrix} \gamma - \zeta & 0 & v \\ 0 & -\gamma I & W^\mathsf{T}\operatorname{diag}(\tau) \\ v^\mathsf{T} & \operatorname{diag}(\tau)W & -2\operatorname{diag}(\tau) \end{bmatrix} \begin{bmatrix} 1 \\ \Delta x \\ \Delta z \end{bmatrix} \le 0$$

*which leads to*

$$\gamma(1 - \|\Delta x\|^2) + 2v\Delta z - \zeta + \begin{bmatrix} \Delta x \\ \Delta z \end{bmatrix}^\mathsf{T} \begin{bmatrix} 0 & W^\mathsf{T}\operatorname{diag}(\tau) \\ \operatorname{diag}(\tau)W & -2\operatorname{diag}(\tau) \end{bmatrix} \begin{bmatrix} \Delta x \\ \Delta z \end{bmatrix} \le 0$$

*Under the constraints in (23), we have $\|\Delta x\| \le 1$ and*

$$\begin{bmatrix} \Delta x \\ \Delta z \end{bmatrix}^\mathsf{T} \begin{bmatrix} 0 & W^\mathsf{T}\operatorname{diag}(\tau) \\ \operatorname{diag}(\tau)W & -2\operatorname{diag}(\tau) \end{bmatrix} \begin{bmatrix} \Delta x \\ \Delta z \end{bmatrix} \ge 0.$$

*Therefore, we must have $v\Delta z \le \frac{\zeta}{2}$ and $\frac{\zeta}{2}$ becomes an upper bound of $L^u$. This leads to the conclusion that $f$ is $\frac{\zeta}{2}$-Lipschitz. The factor of 2 is due to $v\Delta z + \Delta z^\mathsf{T} v^\mathsf{T} = 2v\Delta$ for the scalar output case.*

### E.2 On the Equivalence of SDP Forms (3) and (6)

In this section, we establish the equivalence between (3) and (6). Formally, we have the following result.

**Proposition 2.** *Denote the optimal values of (3) and (6) as $\zeta^*$ and $\zeta^\ddagger$, respectively. Then we have $\zeta^* = \zeta^\ddagger$.*

*Proof.* First, we will prove $\zeta^* \geq \zeta^\ddagger$. Suppose the optimal feasible point for (3) is given by $(\zeta^*, \tau^*, \gamma^*)$. Then we can choose $\zeta = \zeta^*$, $\tau = \tau^*$, $\gamma = \gamma^*$, and $\lambda = 0$, which gives a feasible point for (6). Then this feasible point whose value is exactly $\zeta^*$ gives an upper bound for the optimal value of (6). Consequently, we must have $\zeta^* \geq \zeta^\ddagger$.

Next, we will prove $\zeta^* \leq \zeta^\ddagger$. Suppose the optimal feasible point for (6) is given by $(\zeta^\ddagger, \lambda^\ddagger, \tau^\ddagger, \gamma^\ddagger)$. Then we choose $\zeta = \zeta^\ddagger$, $\tau = \tau^\ddagger + \lambda^\ddagger$, and $\gamma = \gamma^\ddagger + \sum_{j=1}^n a_j \lambda_j^\ddagger$, and show this gives a feasible point for (3) which has the value $\zeta^\ddagger$. To show that our choice gives a feasible point for (3), we only need to verify that the following matrix inequality holds for our choice of $(\zeta, \tau, \gamma)$:

$$\begin{bmatrix} \gamma - \zeta & 0 & v \\ 0 & -\gamma I & W^\mathsf{T} \operatorname{diag}(\tau) \\ v^\mathsf{T} & \operatorname{diag}(\tau)W & -2\operatorname{diag}(\tau) \end{bmatrix} \preceq 0 \tag{24}$$

To verify (24), we can perform the following calculation:

$$\begin{bmatrix} \gamma - \zeta & 0 & v \\ 0 & -\gamma I & W^\mathsf{T} \operatorname{diag}(\tau) \\ v^\mathsf{T} & \operatorname{diag}(\tau)W & -2\operatorname{diag}(\tau) \end{bmatrix}$$
$$= \begin{bmatrix} \gamma^\ddagger + \sum_{j=1}^n a_j \lambda_j^\ddagger - \zeta^\ddagger & 0 & v \\ 0 & -(\gamma^\ddagger + \sum_{j=1}^n a_j \lambda_j^\ddagger)I & W^\mathsf{T} \operatorname{diag}(\tau^\ddagger + \lambda^\ddagger) \\ v^\mathsf{T} & \operatorname{diag}(\tau^\ddagger + \lambda^\ddagger)W & -2\operatorname{diag}(\tau^\ddagger + \lambda^\ddagger) \end{bmatrix}$$
$$= \begin{bmatrix} \gamma^\ddagger + \sum_{j=1}^n a_j \lambda_j^\ddagger - \zeta^\ddagger & 0 & v \\ 0 & -\gamma^\ddagger I & W^\mathsf{T} \operatorname{diag}(\tau^\ddagger) \\ v^\mathsf{T} & \operatorname{diag}(\tau^\ddagger)W & -2\operatorname{diag}(\tau^\ddagger) - \operatorname{diag}(\lambda^\ddagger) \end{bmatrix} + \begin{bmatrix} 0 & 0 & 0 \\ 0 & -(\sum_{j=1}^n a_j \lambda_j^\ddagger)I & W^\mathsf{T} \operatorname{diag}(\lambda^\ddagger) \\ 0 & \operatorname{diag}(\lambda^\ddagger)W & -\operatorname{diag}(\lambda^\ddagger) \end{bmatrix}$$

where the first term is known to be negative semidefinite due to the fact that $(\zeta^\ddagger, \lambda^\ddagger, \tau^\ddagger, \gamma^\ddagger)$ gives a feasible point for (6). Therefore, (24) holds as desired if we can show that the second term is also negative semidefinite. By Schur complement, this is equivalent to verifying

$$-(\sum_{j=1}^n a_j \lambda_j^\ddagger)I + W^\mathsf{T} \operatorname{diag}(\lambda^\ddagger)W \preceq 0$$

Since we have $W^\mathsf{T} \operatorname{diag}(\lambda^\ddagger)W = \sum_{j=1}^n \lambda_j^\ddagger W_j^\mathsf{T} W_j$, we only need to verify

$$-(\sum_{j=1}^n a_j \lambda_j^\ddagger)I + \sum_{j=1}^n \lambda_j^\ddagger W_j^\mathsf{T} W_j \preceq 0$$

Since $a_j = \|W_j\|^2$, we do have $\lambda_j^\ddagger W_j^\mathsf{T} W_j \preceq \lambda_j^\ddagger a_j I$ for all $j$. This leads to the desired conclusion that (24) holds. Therefore, we have obtained a feasible point for (3) which gives the value $\zeta^\ddagger$, and the optimal value of (3) can only be smaller than or equal to $\zeta^\ddagger$. This proves $\zeta^* \leq \zeta^\ddagger$.

To summarize, we must have $\zeta^* = \zeta^\ddagger$, and hence (3) and (6) are equivalent. $\square$

## F Extending Our Results to the Vector Output Case

In this section, we discuss how to extend EP-LipSDP for the scalar output case to the vector output case. The extension is actually quite straightforward. First, we consider the single-layer case. Consider $f(x) = V\sigma(Wx + b^0) + b^1$, where $x \in \mathbb{R}^{n_x}$, $W \in \mathbb{R}^{n \times n_x}$, $b^0 \in \mathbb{R}^n$, $v \in \mathbb{R}^{m \times n}$, $b^1 \in \mathbb{R}^m$, and $f(x) \in \mathbb{R}^m$. The neural network $f : \mathbb{R}^{n_x} \to \mathbb{R}^m$, has $n$ neurons, and $\sigma$ is assumed to be

slope-restricted on $[0, 1]$. In this primal domain, the following QCQP provides the Lipschitz upper bound for $f$:

$$
\begin{aligned}
\max \quad & \|V\Delta z\|^2 \\
\text{subject to} \quad & (\Delta z_i - W_i \Delta x)\Delta z_i \leq 0, \ \forall i \in [n] \\
& \|\Delta x\| \leq 1.
\end{aligned}
\tag{25}
$$

Obviously, Shor's relaxation of the above QCQP is the dual program of the original LipSDP (for the vector output setting). Similar to our previous development, we can introduce redundant trace constraints to (25) and obtain

$$
\begin{aligned}
\max \quad & \|V\Delta z\|^2 \\
\text{subject to} \quad & (\Delta z_i - W_i \Delta x)\Delta z_i \leq 0, \ \forall i \in [n] \\
& \|\Delta x\| \leq 1 \\
& \Delta z_i^2 \leq \|W_i\|_{\text{op}}^2, \forall i \in [n].
\end{aligned}
\tag{26}
$$

Then it is straightforward to verify the dual of the Shor's SDP relaxation of (26) reads as

$$
\begin{aligned}
\min_{\zeta, \lambda, \tau, \gamma} \quad & \zeta \\
\text{subject to} \quad & \begin{bmatrix} \sum_{j=1}^n a_j \lambda_j + \gamma - \zeta & 0 & 0 \\ 0 & -\gamma I & W^\mathsf{T} \operatorname{diag}(\tau) \\ 0 & \operatorname{diag}(\tau)W & -2\operatorname{diag}(\tau) - \operatorname{diag}(\lambda) + V^\mathsf{T}V \end{bmatrix} \preceq 0, \\
& \zeta \geq 0, \ \lambda \geq 0, \ \tau \geq 0, \ \gamma \geq 0.
\end{aligned}
\tag{27}
$$

Since the dual program of the above SDP has an explicit trace constraint, the penalty formulation of the above SDP will be exact. We define

$$
C := \begin{bmatrix} \sum_{j=1}^n a_j \lambda_j + \gamma - \zeta & 0 & 0 \\ 0 & -\gamma I & W^\mathsf{T} \operatorname{diag}(\tau) \\ 0 & \operatorname{diag}(\tau)W & -2\operatorname{diag}(\tau) - \operatorname{diag}(\lambda) + V^\mathsf{T}V \end{bmatrix}.
$$

Then we have the following EP-LipSDP for the vector output case:

$$
\min_{\zeta \geq 0, \lambda \geq 0, \tau \geq 0, \gamma \geq 0} \zeta + \left(2 + \sum_{j=1}^n a_j\right)\lambda_{\max}^+(C).
\tag{28}
$$

The penalty parameter is set as $2 + \sum_{j=1}^n a_j$, which aligns well with our previous developments. We can use the same argument that we developed previously to show that the above EP-LipSDP form has exactly the same optimal value as the original LipSDP.

By slightly modifying the above argument, we can also obtain EP-LipSDP for a multi-layer vector-output network. Specifically, one just removes $v$ and $v^\mathsf{T}$ from the matrix $C$ and then add $V^\mathsf{T}V$ to the last block of $C$. Then the rest of the EP-LipSDP formulation will hold exactly the same as the scalar output case.

### F.1 INITIALIZATION FOR THE VECTOR OUTPUT CASE

Recall the SDP for the vector output of a two-layer network is

$$
\begin{aligned}
\min_{\zeta, \lambda, \tau, \gamma} \quad & \zeta \\
\text{subject to} \quad & \begin{bmatrix} \sum_{j=1}^n a_j \lambda_j + \gamma - \zeta & 0 & 0 \\ 0 & -\gamma I & W^\mathsf{T} \operatorname{diag}(\tau) \\ 0 & \operatorname{diag}(\tau)W & -2\operatorname{diag}(\tau) - \operatorname{diag}(\lambda) + V^\mathsf{T}V \end{bmatrix} \preceq 0, \\
& \zeta \geq 0, \ \lambda \geq 0, \ \tau \geq 0, \ \gamma \geq 0.
\end{aligned}
\tag{29}
$$

We can choose $\lambda_j = 0$, and then we only need $\gamma - \zeta \leq 0$ and $\begin{bmatrix} -\gamma I & W^\mathsf{T} \operatorname{diag}(\tau) \\ \operatorname{diag}(\tau)W & -2\operatorname{diag}(\tau) + V^\mathsf{T}V \end{bmatrix} \preceq 0$. By Schur's lemma, we only need: $\gamma > 0$ and

$$
2\operatorname{diag}(\tau) - V^\mathsf{T}V - \frac{1}{\gamma}\operatorname{diag}(\tau)WW^\mathsf{T}\operatorname{diag}(\tau) \succeq 0.
$$

Let $\mathrm{diag}(\tau) = aI$, then we want

$$2aI - V^\mathsf{T}V - \frac{a^2}{\gamma}WW^\mathsf{T} \succeq 0.$$

We can choose $a = \|V\|_{\mathrm{op}}$ and $\gamma = \|V\|_{\mathrm{op}}^2 \|W\|_{\mathrm{op}}^2$, and $\zeta = \|V\|_{\mathrm{op}}^2 \|W\|_{\mathrm{op}}^2$. Because we will square root $\zeta$ to estimate the Lipschitz constant, this recovers the naive upper bound of the Lipschitz constant: $\|V\|_{\mathrm{op}} \|W\|_{\mathrm{op}}$.

For 3-layer network, the SDP constraint is:

$$C = \begin{bmatrix} \gamma - \zeta & 0 & 0 & 0 \\ 0 & -\gamma I & W_1^\mathsf{T}\Lambda_1 & 0 \\ 0 & \Lambda_1 W_1 & -2\Lambda_1 & W_2\Lambda_2 \\ 0 & 0 & \Lambda_2 W_2 & -2\Lambda_2 + V^\mathsf{T}V \end{bmatrix} \preceq 0$$

Let

$$C_1 = \begin{bmatrix} \gamma - \zeta & 0 & 0 & 0 \\ 0 & 0 & 0 & 0 \\ 0 & 0 & 0 & 0 \\ 0 & 0 & 0 & 0 \end{bmatrix},$$

$$C_2 = \begin{bmatrix} 0 & 0 & 0 & 0 \\ 0 & -\gamma I & W_1^\mathsf{T}\Lambda_1 & 0 \\ 0 & \Lambda_1 W_1 & -\Lambda_1 & 0 \\ 0 & 0 & 0 & 0 \end{bmatrix},$$

and

$$C_3 = \begin{bmatrix} 0 & 0 & 0 & 0 \\ 0 & 0 & 0 & 0 \\ 0 & 0 & -\Lambda_1 & W_2\Lambda_2 \\ 0 & 0 & \Lambda_2 W_2 & -2\Lambda_2 + V^\mathsf{T}V \end{bmatrix}.$$

Clearly, $C = C_1 + C_2 + C_3$, then we only need $C_1, C_2, C_3 \preceq 0$.

We can have $\Lambda_1 = aI$, $\Lambda_2 = bI$. To have $C_3 \preceq 0$, then by the Schur complement lemma, we need

$$2bI - V^\mathsf{T}V - \frac{b^2}{a}W_2 W_2^\mathsf{T} \succeq 0.$$

We can have $b = \|V\|_{\mathrm{op}}^2$ and $a = \|V\|_{\mathrm{op}}^2 \|W_2\|_{\mathrm{op}}^2$

To have $C_2 \preceq 0$, we need

$$aI - \frac{a^2}{\gamma}W_1 W_1^\mathsf{T} \succeq 0.$$

We can have $\gamma = a \|W_1\|_{\mathrm{op}}^2 = \|V\|_{\mathrm{op}}^2 \|W_2\|_{\mathrm{op}}^2 \|W_1\|_{\mathrm{op}}^2$.

In general, for a multi-layer network, we can apply the above argument recursively:

$$\begin{bmatrix} \gamma - \zeta & 0 & 0 & \cdots & 0 \\ 0 & -\gamma I & W^{(1)}\Lambda_1 & \cdots & 0 \\ \vdots & \cdots & \cdots & \ddots & \vdots \\ 0 & 0 & \cdots & \Lambda_{d-1}(W^{d-1}) & -2\Lambda_{d-1} + V^\mathsf{T}V \end{bmatrix}.$$

We will have $\gamma = \zeta = \Pi_{i=1}^{d-1} \|W_i\|_{\mathrm{op}}^2 \|V\|_{\mathrm{op}}^2$ and $\Lambda_i = \Pi_{j=i+1}^{d-1} \|W_j\|_{\mathrm{op}}^2 \|V\|_{\mathrm{op}}^2 \cdot I$.

Table 3: In this table, we present results of comparing different variants of our approach to LipSDP on a few MNIST DNNs. Among all the results, we can see that if we use the exact eigenvalue computation instead of Lanczos approximation, LipDiff results in very close values to MOSEK's results on all three DNN architectures.

| Datasets | Models | | Product | LipSDP-MOSEK | LipSDP-SCS | LipDiff | LipDiff-Ex | LipDiff-Dense | LipDiff-Rand | LB |
|---|---|---|---|---|---|---|---|---|---|---|
| **MNIST** | **DNN** | Result | 9.31 | 4.82 | 4.82 | 4.90 | 4.86 | 4.96 | 5.89 | 1.57 |
| | | Time (s) | 0.13 | 54.57 | 2706.5 | 28.69 | 19.27 | 12.48 | 29.27 | 4.86 |
| | | Memory (MB) | 1.54 | 169.83 | 256.46 | 118 | 114 | 114 | 118 | 2134 |
| **MNIST** | **WIDE** | Result | 9.27 | 4.87 | 4.87 | 4.93 | 4.89 | 4.93 | 5.57 | 1.51 |
| | | Time (s) | 0.10 | 214.67 | 5014.93 | 65.89 | 41.85 | 29.03 | 64.10 | 4.88 |
| | | Memory (MB) | 3.23 | 297.68 | 415.90 | 230 | 192 | 192 | 230 | 2424 |
| **MNIST** | **DNN3** | Result | 17.29 | 7.43 | 7.38 | 7.76 | 7.51 | 7.60 | 11.47 | 2.03 |
| | | Time (s) | 0.12 | 649.40 | 13736.16 | 238.40 | 128.50 | 83.55 | 241.34 | 4.95 |
| | | Memory (GB) | 3.23 | 585.60 | 810.26 | 362 | 284 | 284 | 362 | 2622 |

## G ADDITIONAL EXPERIMENTAL RESULTS

We provide more experiments on two additional architectures on MNIST that the LipSDP-MOSEK solver can handle: 1. A single hidden layer with $512$ hidden nodes (denoted as *WIDE*); 2. a two-hidden-layer network (denoted as *DNN3*), and each of the hidden layers has $512$ ReLU nodes.

Additionally, we provide two new baselines: LB and LipSDP-SCS. For LB, we randomly sampled $500,000$ points from the input space and computed the maximum $\ell_2$ norm of all the gradients induced on these samples, which serve as a lower bound. For SCS, we used the default SCS solver provided by CVXPY to solve the SDP program, with max iterations of $2,500$. Notice that SCS solver cannot scale to the MNIST-CNN network, similar to the MOSEK solver. The specification of *MNIST-DNN* is included in the main paper.

The results are summarized in Table 3.

Notice that we do not know how good the lower bound is because the number of samples is unlikely to be sufficient. For example, even if we only consider the vertices of the input on the [0,1]-hypercube, there would be $2^{784}$ vertices for MNIST and $2^{3072}$ vertices for CIFAR10, which are much greater than 500,000.

For SCS, we notice that if we set the max iterations too small (for example, $50$), the result can be very unstable. It can return either 0 or inf, which is not useful at all. On MNIST-DNN3, SCS provided a value smaller than the MOSEK solver, which we do not know whether it is a valid upper bound for the Lipschitz constant. However, LipDiff always returns a valid upper bound.

On the other hand, because the network is relatively small compared to CNNs, computing exact eigenvalue and using the dense representation of the matrix is more efficient than the sparse representation. The results from LipDiff-Ex and LipDiff-Dense are more desirable. However, because the main goal of LipDiff is to scale LipSDP to huge networks that the general SDP solver cannot handle at all, sparse representation and the Lanczos approximation would be more appropriate for these cases, as shown in Table 2.

## H ALEXNET EXPERIMENTAL RESULT

AlexNet consists of two parts: a feature extraction sub-network and a classifier sub-network. Because the operator norm is sub-multiplicative, we apply the scalar LipDiff on the classifier, and also the vector output version of LipDiff on the feature extraction network. More specifically, the feature extraction network contains three separated convolutional parts, and we apply LipDiff on the final convolutional part, which consists of three consecutive convolutional layers, and all of them have $3 \times 3$ kernels and padding 1. For the classifier sub-network, it contains 3 fully connected layers.

For the classifier, we again choose class $8$ as in the evaluations of MNIST and CIFAR10. If we expand the matrix of the classifier, the SDP has a size of $17\,409 \times 17\,409$. The matrix norm product bound of the classifier is $67.77$, while LipDiff produces the Lipschitz constant of $22.41$, which leads to a $66.9\%$ decrease.

For the three convolutional layers, if we expand the matrix of the classifier, the SDP has a size of $140\,609 \times 140\,609$. The matrix norm product bound of the classifier is $2886.73$, while LipDiff

Table 4: We summarize AlexNet experimental results: we run scalar-LipDiff on the classifier part, and vector-LipDiff on a sub-network of the feature extractor. On the classifier, LipDiff achieves a **66.9**% improvement on the Lipschitz constant estimation; and on the feature extractor, LipDiff decreases the Lipschitz constant by **18.4**%, compared to the matrix-norm bound. Combining these two results, LipDiff achieves a total of **73**% improvement for AlexNet Lipschitz constant estimation compared to the matrix norm product bound.

| Model | Structure | SDP size | Matrix Norm Bound | LipDiff | Improvement |
|---|---|---|---|---|---|
| classifier | 3FC | $17\,409 \times 17\,409$ | 67.77 | 22.41 | 66.9% |
| feature extractor | 3C | $140\,609 \times 140\,609$ | 2886.73 | 2354.77 | 18.4% |

produces the Lipschitz constant of 2354.77, which leads to a $18.4\%$ decrease. In this experiment, because the matrix is very large, we are unable to get the exact eigenvalue, instead we used Lanczos algorithm to estimate the eigenvalue with Lanczos steps 9200. The initialization for the vector ex-LipSDP comes from appendix F.1.

Combining the results from both parts, LipDiff reduces the Lipschitz constant of AlexNet by $73\%$ compared to the naive matrix norm product bound.

# I  FURTHER DISCUSSIONS AND EXPLANATIONS

## I.1  TECHNICAL NOVELTY

In this section, we briefly summarize the technical novelty of our work.

**Adding redundant trace constraint**: To obtain the exact penalty form of LipSDP, we need to add an explicit redundant trace constraint to the primal form of LipSDP. How to add this explicit trace constraint highly depends on the network structure, and there is no general recipe for doing this. We delicately exploit the structure of the neural network to add the redundant trace constraint in a layer-by-layer manner, leading to the first exact penalty form of LipSDP.

**Proof techniques beyond Liao et al. (2023) and Ding & Grimmer (2023)**: Our exact penalized SDP formulation is motivated by the recent advances on first-order methods for convex nonsmooth optimization (Liao et al., 2023; Ding & Grimmer, 2023). However, our result is not simply a direct application of their results. In particular, all the existing results from Liao et al. (2023) and Ding & Grimmer (2023) require the penalty parameter to be strictly larger than the trace bound of the semidefinite variable, while our main technical results in Theorem 1 and Theorem 2 allow for the non-strict inequality, i.e., our penalty parameter only needs to be larger than or equal to the trace bound. Also, our proofs for Theorem 1 and Theorem 2 directly exploit the neural network structure, and consequently our argument is simpler than those in Liao et al. (2023) and Ding & Grimmer (2023). Our proof is also self-contained, while those of Liao et al. (2023); Ding & Grimmer (2023) reply on other technical results, such as strong duality and a general penalized result in Ruszczynski (2011, Theorem 7.21).

**Network-dependent tricks for making our algorithm LipDiff work on practical problems**: In Section 5, we provide further network-dependent tricks for making our method LipDiff work on practical problems. Our tricks include analytically deriving an initialization with nontrivial matrix analysis (see Appendix B) and utilizing the sparse structure of the constraint to faster approximate the maximum eigenvalue. This analytical initialization guarantees that our method is always as good as the matrix norm product (which is currently the state-of-the-art technique for estimating Lipschitz bounds of large neural networks) and makes LipDiff practical. From our empirical evaluation, for large networks (MNIST/CIFAR10-CNNs), if we randomly initialize the variables, it is very hard for LipDiff to find a good solution. We propose to initialize from a feasible point that exactly corresponds to the matrix norm product bound, and we leverage matrix analysis tools to derive the analytical form of this initial point. Such a development depends on the network structure, and is completely a new contribution.

## I.2    POSSIBILITY OF TIGHTENING LipSDP

To the best of our knowledge, LipSDP is recognized as the method that can give the least conservative $\ell_2$-Lipschitz upper bound with polynomial time guarantees. On small-scale problems where LipSDP can be efficiently solved, numerically it has been very difficult to find a polynomial-time algorithm that can give less conservative $\ell_2$-Lipschitz upper bounds than LipSDP. Therefore, it is reasonable to focus on how to make LipSDP scalable and memory-efficient. How to tighten LipSDP is beyond the scope of this paper. Tightening LipSDP is an interesting problem and hence should be pursued in the future. In the meantime, given the current redundant trace constraint, tuning the parameter $\rho$ may not be the right way to tighten LipSDP. Specifically, we have theoretically proved that LipSDP with or without the redundant trace constraint (i.e. (3) and (6) in our paper) actually give the same solution (see Appendix E for a detailed proof). It is true that adding redundant constraints can refine the solutions for many SDP relaxations. However, for LipSDP, it is quite unfortunate that the current redundant trace constraint does not refine the solution quality. Consequently, given the current redundant trace constraint, tuning $\rho$ does not seem to be the plausible way to tighten LipSDP. We agree that it is possible to develop other redundant constraints that may be incorporated into the original QCQP formulation to tighten LipSDP. This will be a future direction for us.

## I.3    RELATION BETWEEN SDPs IN LIPSCHITZ ESTIMATION AND LOCAL ROBUSTNESS VERIFICATION

There are recent advances in SDP formulation for the robustness verification of neural networks. One of the early formulations appeared in Raghunathan et al. (2018). Indeed, the high-level ideas in both robustness verification and Lipschitz constant estimation are very similar: both of them formulate the problems into a QCQP form and then relax the QCQP using standard Lagrangian relaxation or Shor'r relaxation. In principle, if one can formulate a better QCQP form (i.e., adding some effective redundant constraints, see e.g., Batten et al. (2021)), the resulting SDP formulation would provide a better estimation. Some tightness analysis was carried out in Zhang (2020), however, the assumptions therein are very restrictive, which may not hold for practical neural networks. The paper by Dathathri et al. (2020) aims to develop a customized first-order solver to improve the scalability of the SDP in Raghunathan et al. (2018). Our work is also partially motivated by Dathathri et al. (2020). We note that the SDP formulations for the robustness verification of neural networks and Lipshcitz estimation are indeed very different. The line of work on robustness verification may give new insights, but it is nontrivial to directly apply those advances. For example, we have proved that adding some redundant constraints in our formulation does not improve the tightness of the resulting SDP formulation. It is a promising direction to unify the QCQP and SDP formulations and their solutions for both robustness verification and Lipchitz constant estimation.

## I.4    RELATED WORK ON POLYNOMIAL OPTIMIZATION

Mai et al. (2022) considers a constant trace property in polynomial optimization. In particular, the variable $X$ satisfies $\text{trace}(X) = c$, which is a constant. Then, Mai et al. (2022) translate this constant trace formulation into an eigenvalue problem via a standard procedure in Helmberg & Kiwiel (2002). However, this property does not hold in LipSDP arising from the Lipschitz constant estimation of neural networks. Indeed, our first theoretical guarantees in Theorem 1 and Theorem 2 offer new exact penalty SDP formulations that are suitable for the application of first-order subgradient methods. Our penalized SDP formulation is motivated by the recent advances in convex nonsmooth optimization (Liao et al., 2023; Ding & Grimmer, 2023). We exploit the problem structure (especially the neural network structure) to provide a simple elegant proof of our main theoretical guarantees in Theorem 1 and Theorem 2.

## I.5    IPM VS FIRST-ORDER METHODS

The per iteration computation of first-order methods is much cheaper than IPM, while IPM requires much less iterations in total to achieve an $\epsilon$-solution. To achieve an $\epsilon$-approximate solution, the total iteration number required by IPM is on the order of $O(\log(1/\epsilon))$. For LipDiff, we are applying a subgradient method to solve a convex nonsmooth optimization problem, and the total iteration number needed to get an $\epsilon$ solution is on the order of $O(1/\epsilon^2)$. However, for each iteration, the computational/memory efficiency for IPM is much worse than our first-order subgradient method.

In practice, Lipschitz estimation of neural networks typically does not require using very small $\epsilon$, i.e. $\epsilon$ is typically set to be on the order of $0.1$ or at most $0.01$. Therefore, we argue that per iteration complexity/efficiency matters more for the Lipschitz estimation problem of larger neural networks. In addition, we have developed the special analytical initialization to reduce the iteration number needed by LipDiff (see Appendix B). This justifies the significance of our contribution.

## I.6 COMPARISON WITH COSMO

We have tried COSMO in some additional experiments, and it did not work well. The COSMO solver is implemented in Julia and may be called by the JuMP optimization modeling framework. We implemented LipSDP in Julia with the COSMO solver which gave an accurate solution (identical to MOSEK) for small networks with SDP dim $10 \times 10$. Unfortunately, for small MNIST networks of just two layers (resulting in an SDP of dim $800 \times 800$), COSMO appears to be unstable and will not converge. Finally, we want to comment that COSMO aims to solve the KKT condition of conic programs, and does not guarantee the returned solution to be valid Lipschitz upper bounds. In contrast, our proposed first-order method LipDiff (EP-LipSDP) guarantees that any value returned by LipDiff is always a valid upper bound higher than or equal to LipSDP's optimal value, because the iterations of LipDiff naturally correspond to feasible points of LipSDP which directly gives Lipschitz upper bounds.

## I.7 CONVEX NONSMOOTH OPTIMIZATION VS. MINIMAX FORMULATION

Here we clarify that we treat EP-LipSDP as a convex nonsmooth optimization problem instead of a minimax problem. For simplicity, we restate the EP-LipSDP problem here as

$$\min_{\zeta \geq 0, \lambda \geq 0, \tau \geq 0, \gamma \geq 0} J(\zeta, \lambda, \tau, \gamma) := \zeta + \Big(2 + \sum_{j=1}^{n} a_j\Big)\lambda_{\max}^+(C(\zeta, \lambda, \tau, \gamma)).$$

where $C(\zeta, \lambda, \tau, \gamma)$ is defined as

$$C(\zeta, \lambda, \tau, \gamma) := \begin{bmatrix} \sum_{j=1}^{n} a_j\lambda_j + \gamma - \zeta & 0 & v \\ 0 & -\gamma I & W^{\mathsf{T}}\operatorname{diag}(\tau) \\ v^{\mathsf{T}} & \operatorname{diag}(\tau)W & -2\operatorname{diag}(\tau) - \operatorname{diag}(\lambda) \end{bmatrix}.$$

Since $C$ is linear in $(\zeta, \lambda, \tau, \gamma)$, one can show that $J$ is a convex nonsmooth function of $(\zeta, \lambda, \tau, \gamma)$. The nonsmoothness is introduced by the operation $\lambda_{\max}^+$. We emphasize that $J$ is a function of $(\zeta, \lambda, \tau, \gamma)$. Although $J$ is nonsmooth, we can still apply the subgradient method with box projection to solve the above problem. Notice that the operation $\lambda_{\max}^+$ is not performing a maximization over $(\zeta, \lambda, \tau, \gamma)$, and hence we are not solving a minimax problem.

