# OpenReview forum: "On the Scalability and Memory Efficiency of Semidefinite Programs  for Lipschitz Constant Estimation of Neural Networks"
_ICLR.cc/2024/Conference — ICLR 2024 poster_

### Official Review · Reviewer_GVUd · 2023-10-24

**Soundness:** 3 good
**Presentation:** 4 excellent
**Contribution:** 2 fair
**Rating:** 5
**Confidence:** 4

**Summary:**

Papers concerns a new SDP method to approximate the Lipschitz constant of neural networks, with the main benefit being its scalability due to the reduction of explicit constraints as penalizations. The proposed methods is itself useful, however the paper did not convince me of its technical innovation. It just seems to me that the authors simply took the methods from (Liao et al., 2023) and applied it to the LipSDP method to ensure that they have the same optimal solution.

**Strengths:**

The proposed technique itself is nice, and may be beneficial in many aspects. The numerical experiments are also convincing, demonstrating its usability. The presentation is clear and easy to follow.

**Weaknesses:**

1. The theoretical innovation is relatively weak, because anyone can apply different techniques in the convex programming space and use it on LipSDP to claim as a new idea. I didn't see any specific analysis and/or new technique proposed specifically for the Lipschitz problem of NN.  Maybe I am overlooking some details, and if so please point me to these places.


2. The authors only showed that their new formulation has the same optimal value as the old LipSDP problem. However since LipSDP is a relaxation, there are no guarantees of its accuracy. The authors introduced redundant constraints in this paper, and this kind of technique is usually used to tighten relaxation gap in classic literature, so I hope the authors could also touch upon the tightness of relaxation, especially by tuning the parameter $\rho$, as I believe this is possible to analyze.

Typos
- On page 3, it seems like when you say "second term of the left hand, etc" you meant the third term, and you grouped the first two terms together implicitly. Also please explain how your third term ties to the relu function (experts may understand, but since you're explaining here please be more precise).

**Questions:**

Not questions, but I encourage the authors to check a major line of work concerning the SDP verification of robustness of neural networks. That problem and your Lipschitz problem are very similar from a technical standpoint, and there are already a number of different branches in that space. For example brand-and-bound methods, non-convex cut, chordal decomposition (as you have mentioned), and theoretical analysis of relaxation quality. By exploring that problem I think you can benefit the Lipschitz problem as well.

---

> ### Author Response · Authors · 2023-11-22
> **Response to Reviewer GVUd, Part 1**
>
> We appreciate your professional review. Here are our responses to the concerns:
>
> > It just seems to me that the authors simply took the methods from [1] and applied it to the LipSDP method to ensure that they have the same optimal solution.  I didn't see any specific analysis and/or new technique proposed specifically for the Lipschitz problem of NN. Maybe I am overlooking some details, and if so please point me to these places.
>
> As pointed out by Reviewer CCdS, our development is not a simple combination of existing convex optimization techniques. As a matter of facts, our technique does exploit the special structure of the neural network problem in various ways. Now we elaborate on this.
>
> - Adding redundant trace constraint: To obtain the exact penalty form of LipSDP, we need to add an explicit redundant trace constraint to the primal form of LipSDP. How to add this explicit trace constraint highly depends on the network structure, and there is no general recipe for doing this. We delicately exploit the structure of the neural network to add the redundant trace constraint in a layer-by-layer manner, leading to the first exact penalty form of LipSDP.
>
>
> - Proof techniques beyond [Liao'23,Ding'23]: It is true that our exact penalized SDP formulation is motivated by the recent advances on first-order methods for convex nonsmooth optimization [Liao'23, Ding'23]. However, our result is not simply a direct application of their results. In particular, all the existing results from [Liao'23, Ding'23] require the penalty parameter to be strictly larger than the trace bound of the semidefinite variable, while our main technical results in Theorem 1 and Theorem 2 of our paper allow for the non-strict inequality, i.e., our penalty parameter only needs to be larger than or equal to the trace bound. Also, our proofs for Theorem 1 and Theorem 2 directly exploit the neural network structure, and consequently our argument is actually simpler than those in [Liao'23, Ding'23]. Our proof is also self-contained, while those of [Liao'23, Ding'23] reply on other technical results, such as strong duality and a general penalized result in [Ruszczynski'11, Theorem 7.21].
>
>
>
> - Network-dependent tricks for making our algorithm LipDiff work on practical problems:  In Section 5 of our paper, we provide further network-dependent tricks for making our method LipDiff work on practical problems. Our tricks include analytically deriving an initialization with nontrivial matrix analysis (see Appendix B) and utilizing the sparse structure of the constraint to faster approximate the maximum eigenvalue. This analytical initialization guarantees that our method is always as good as the matrix norm product (which is currently the state-of-the-art technique for estimating Lipschitz bounds of large neural networks) and makes LipDiff practical. From our empirical evaluation, for large networks (MNIST/CIFAR10-CNNs), if we randomly initialize the variables, it is very hard for LipDiff to find a good solution. We propose to initialize from a feasible point which exactly corresponds to the matrix norm product bound, and we leverage matrix analysis tools to derive the analytical form of this initial point. Such a development depends on the network structure, and is completely a new contribution.
>
> Therefore, we believe that our work is not a simple application of existing convex optimization methods. We have also included the above discussion in Appendix G.1 of our revised paper.

---

> ### Author Response · Authors · 2023-11-22
> **Response to Reviewer GVUd, Part 2**
>
> > Since LipSDP is a relaxation, there are no guarantees of its accuracy. I hope the authors could also touch upon the tightness of relaxation, especially by tuning the parameter p as I believe this is possible to analyze.
>
> To the best of our knowledge, LipSDP is still recognized as the method that can give the least conservative $\ell_2$-Lipschitz upper bound with polynomial time guarantees.
> On small scale problems where LipSDP can be efficiently solved, numerically it has been very difficult to find a polynomial-time algorithm that can give less conservative $\ell_2$-Lipschitz upper bounds than LipSDP. Therefore, it is very reasonable for our paper to focus on how to make LipSDP scalable and memory-efficient.  We agree with the reviewer that tightening LipSDP is an interesting problem and hence should be pursued in the future. However, with our current redundant trace constraint, tuning the parameter $\rho$ may not be the right way to tighten LipSDP. Specifically, we have theoretically proved that LipSDP with or without the redundant trace constraint (i.e. (3) and (6) in our paper) actually give the same solution (see Appendix E.2 of our revised paper for a detailed proof). It is true that adding redundant constraints can refine the solutions for many SDP relaxations. However, for LipSDP, it is quite unfortunate that the current redundant trace constraint does not refine the solution quality (Appendix E.2 exactly proves this). Consequently, given the current redundant trace constraint, tuning $\rho$ does not seem to be the plausible way to tighten LipSDP. We agree that it is possible to develop other redundant constraints that may be incorporated into the original QCQP formulation to tighten LipSDP. This will be a future direction for us. We have included the above point in Appendix G.2 of our revised paper.
>
>
> > On page 3, it seems like when you say "second term of the left hand, etc" you meant the third term, and you grouped the first two terms together implicitly.
>
> Thanks for pointing this out. Your understanding is correct. We have fixed this typo in our revised version of the paper.
>
> > Also please explain how your third term ties to the relu function (experts may understand, but since you're explaining here please be more precise)
>
> This comes from the slope-restricted interpretation of ReLU function: the slope between any two points on the ReLU function is always between 0 and 1. This was used in the original LipSDP paper [Fazlyab'19, section 2.2] and the follow-up work [Wang'22, section 4]. We provided detailed references in the revised paper.
>
>
> > Not questions, but I encourage the authors to check a major line of work concerning the SDP verification of robustness of neural networks.
>
> We would like to thank the reviewer for this valid suggestion. Indeed, we are aware of the recent advances in SDP formulation for the local robustness verification of neural networks under $\ell_\infty$ perturbations. One of the early formulations appeared in [Raghunathan'18], where SDPs are used to address $\ell_\infty$ perturbations. Indeed, the high-level ideas in both local robustness verification and Lipschitz constant estimation are very related: both of them formulate the problems into a QCQP form and then relax the QCQP using standard Lagrangian relaxation or Shor’s relaxation. In principle, if one can formulate a better QCQP form, the resulting SDP formulation would provide a better estimation [Batten'21;Lan'22]. Some tightness analysis was carried out in [Zhang'20], however, the assumptions therein are very restritive, which may not hold for practical neural networks. The paper by [Dathathri'21] aims to develop a customized first-order solver to improve the scalability of the SDP in [Raghunathan'18]. Our work is also partially motivated by [Dathathri'21], although the techniques are very different. We note that the SDP formulations for the local $\ell_\infty$ robustness verification of neural networks and Lipshcitz estimation are indeed very different. The line of work on local $\ell_\infty$ robustness verification may give new insights, but it is nontrivial to directly apply those advances. For example, we have proved that adding some redundant trace constraints in our formulation does not improve the tightness of the resulting SDP formulation (i.e. in Appendix E.2 of our revise paper, we have proved that (3) and (6) give the same results). We definitely agree that it is a promising direction to unify the QCQP and SDP formulations and their solutions for both robustness verification and Lipchitz constant estimation. We have included the above discussion in Appendix G.3 of our revised paper.

---

> ### Author Response · Authors · 2023-11-22
> **List of References Mentioned in the Above Response**
>
> For your convenience, here we provide the list of the references mentioned in the above response.
>
> [Liao'23]Feng-Yi Liao, Lijun Ding, and Yang Zheng. An overview and comparison of spectral bundle methods for primal and dual semidefinite programs.
>
> [Ding'23]Lijun Ding and Benjamin Grimmer. Revisiting spectral bundle methods: Primal-dual (sub) linear convergence rates. SIAM Journal on Optimization, 2023
>
> [Ruszczynski'11]Andrzej Ruszczynski. Nonlinear optimization. Princeton university press, 2011.
>
> [Raghunathan'18]Aditi Raghunathan, Jacob Steinhardt and Percy Liang. Semidefinite relaxations for certifying robustness to adversarial examples. NeurIPS 2018
>
> [Batten'21]Batten, B., Kouvaros, P., Lomuscio, A., & Zheng, Y. (2021). Efficient neural network verification via layer-based semidefinite relaxations and linear cuts. IJCAI.
>
> [Lan'22]Lan, Jianglin, Yang Zheng, and Alessio Lomuscio. "Tight neural network verification via semidefinite relaxations and linear reformulations." Proceedings of the AAAI Conference on Artificial Intelligence. Vol. 36. No. 7. 2022.
>
> [Zhang'20]Zhang, Richard. "On the tightness of semidefinite relaxations for certifying robustness to adversarial examples." Advances in Neural Information Processing Systems 33 (2020): 3808-3820.
>
> [Dathathri'21]Sumanth Dathathri, Krishnamurthy Dvijotham, Alexey Kurakin, Aditi Raghunathan, Jonathan Uesato, Rudy R Bunel, Shreya Shankar, Jacob Steinhardt, Ian Goodfellow, Percy S Liang, and Pushmeet Kohli. Enabling certification of verification-agnostic networks via memory-efficient semidefinite programming.
>
> [Fazlyab'19]Mahyar Fazlyab, Alexander Robey, Hamed Hassani, Manfred Morari, and George Pappas. Efficient and accurate estimation of lipschitz constants for deep neural networks
>
> [Wang'22]Zi Wang, Gautam Prakriya, and Somesh Jha. A quantitative geometric approach to neural-network smoothness. NeurIPS 2022

---

### Official Review · Reviewer_peb1 · 2023-10-25

**Soundness:** 3 good
**Presentation:** 3 good
**Contribution:** 2 fair
**Rating:** 6
**Confidence:** 4

**Summary:**

The authors consider the classical Shor semidefinite programming (SDP) relaxation of the Lipschitz constant of neural networks.
They formulate this SDP program as a non-smooth eigenvalue problem with an exact penalty (EP) parameter. The resulting formulation is called EP-LipSDP.
The exact penalty parameter is obtained by means of proper redundant quadratic constraints.

Then they provide different numerical algorithms to handle this eigenvalue optimization, including eigenvector approximation via Lancsoz' method, sparse matrix multiplication, and analytical initialization. These different algorithms allow one to tune the running time and convergence of the optimization process.

The resulting algorithm, called LipDiff is compared to standard techniques on three neural network benchmarks: MNIST DNN, MNIST CNN and CIFAR10 CNN.

**Strengths:**

The authors can obtain a non-trivial Lipschitz constant estimate on the CIFAR10 CNN.
Thanks to the different numerical techniques, the resulting algorithm can be tuned for adaptation to allocated computational resources, in particular memory.

**Weaknesses:**

1. The initial problem of Lipschitz constant computation can actually be formulated as a polynomial optimization problem. The authors have almost completely forgotten to mention related papers based on techniques from this field (at the exception of the one by Latorre et al.), in particular the following ones:

- Chen et al. Semialgebraic optimization for Lipschitz constants of ReLU networks
- Chen et al. Semialgebraic representation of monotone deep equilibrium models and applications to certification
- Mai et al. Exploiting constant trace property in large-scale polynomial optimization

The latter paper relies on the constant (rather than bounded) trace property, and a proper comparison would be valuable.


2. The LipDiff algorithm might not be very accurate in general. From the comparison on the basic benchmark MNIST DNN, one notices that LipSDP obtains a better bound. It would be interesting to see how LipSDP behaves on, e.g., MNIST CNN by using LipSDP with Mosek on a scalable computing platform with more RAM.
First order methods are usually much faster than second-order solvers (in particular interior-point methods) but they are also much less accurate in general. The trade-off between accuracy and efficiency should be more explicitly investigated.
It would be also interesting to see what kind of results first-order SDP solvers like COSMO would yield.

**Questions:**

- p2: "Dathathri et al. (2020) solved a different problem" => which problem?

- p7: Typo "the the"

---

> ### Author Response · Authors · 2023-11-22
> **Response to Reviewer peb1, Part 1**
>
> Thanks for your detailed review and useful feedback.  Here we address all your comments.
>
> > The initial problem of Lipschitz constant computation can actually be formulated as a polynomial optimization problem. The authors have almost completely forgotten to mention related papers based on techniques from this field (at the exception of the one by Latorre et al.).
>
> Thanks for this valuable comment. We have corrected the typo and added related work on polynomial optimization techniques in the revised paper.
>
>
> > The latter paper relies on the constant (rather than bounded) trace property, and a proper comparison would be valuable.
>
> The paper by Mai et al. considers a constant trace property in polynomial optimization. In particular, the decision variable X satisfies trace(X) = c, which is a constant. Then, the authors translate this constant trace formulation into an eigenvalue problem via a standard procedure in [Helmberg'00]. However, this property does not hold in LipSDP arising from the Lipschitz constant estimation of neural networks. Indeed, our theoretical guarantees in Theorem 1 and Theorem 2 offers a new penalized SDP formulation that is suitable for the application of first-order subgradient methods. Our penalized SDP formulation is motivated by the recent advances in convex nonsmooth optimization (cf. [Liao'23], and [Ding'23]). We exploit the problem structure (especially the neural network structure) to provide a simple elegant proof of our main theoretical guarantees in Theorem 1 and Theorem 2. We have added Appendix G.4 to include the above discussion.
>
> > From the comparison on the basic benchmark MNIST DNN, one notices that LipSDP obtains a better bound.
>
>
> We have conducted addtional experiments (see the general response section). Among all three DNN evaluations, the gap between LipSDP and our approach is actually small.
>
>
>
> >  It would be interesting to see how LipSDP behaves on, e.g., MNIST CNN by using LipSDP with Mosek on a scalable computing platform with more RAM.
>
> We sincerely value this useful comment.
> We have added new experiments on Larger MNIST DNN (architectures larger than the MNIST-DNN evaluated in our original paper) to compare LipDiff with LipSDP. Please see the numerical results provided in our general response (and also added in Appendix F of our revised paper). We tried to run LipSDP with Mosek on MNIST CNN, and our current computing resources are not enough to solve the problem (our experiments are run on a server with thirty-two AMD processors, 528 GB of memory, and four Nvidia A100 GPUs, each with 80 GB of memory).
> It seems quite challenging to solve LipSDP with Mosek on MNIST CNN without resorting to other relaxation tricks which explicitly exploit the structures of CNNs. We hope that our additional experiments on Larger MNIST DNN with LipSDP (Mosek) can give the reviewer a better idea of how to compare LipSDP with LipDiff.
>
>
>
>
>
> > First order methods are usually much faster than second-order solvers (in particular interior-point methods) but they are also much less accurate in general. The trade-off between accuracy and efficiency should be more explicitly investigated.
>
>
> Thanks for this great comment. Now we discuss the trade-off between accuracy and efficiency for the interior-point methods (IPM) and the first-order methods. The per iteration computation of first-order methods is much cheaper than IPM, while IPM requires much less iterations in total to achieve an $\epsilon$-solution. To achieve an $\epsilon$-approximate solution, the total iteration number required by IPM is on the order of $O(\log(1/\epsilon))$. For LipDiff, we are applying a subgradient method to solve a convex nonsmooth optimization problem, and the total iteration number needed to get an $\epsilon$ solution is on the order of $O(1/\epsilon^2)$.
> However, for each iteration, the computational/memory efficiency for IPM is much worse than our first-order subgradient method.
> Let $n$ be the number of nodes in the network, and $p$ be the number of parameters in the network. In general, $n< p< n^2$ and if the network is deep, $p<< n^2$. The per iteration memory complexity for IPM is $O(n^4)$ and computational complexity is $O(n^6)$. While for our method (LipDiff), the per iteration memory complexity is $O(L p)$, and per iteration computational complexity is $O(L p)$, where $L$ is the number of Lanczos iterations used for approximate the maximum eigenvalue. In our evaluation, $L$ ranges from $15$ to $50$. In practice, Lipschitz estimation of neural networks typically does not require using very small $\epsilon$,i.e. $\epsilon$ is typically set to be on the order of $0.1$ or at most $0.01$. Hence we argue that per iteration complexity/efficiency matters more for the Lipschitz estimation problem of larger neural networks. Our analytical initialization further reduces the iteration number for LipDiff. This justifies the significance of our method. We added our response as Appendix G.5 of our revised paper.

---

> > ### Comment · Reviewer_peb1 · 2023-11-22
> > **Answer to authors' response**
> >
> > Thanks for these additional experiments and clarifications. I will maintain my score.

---

> ### Author Response · Authors · 2023-11-22
> **General Response to Reviewer peb1, Part 2**
>
> > It would be also interesting to see what kind of results first-order SDP solvers like COSMO would yield.
>
> Thanks for this comment. We have tried COSMO on MNIST, and it did not work well. The COSMO solver is implemented in Julia and may be called by the JuMP optimization modeling framework. We implemented LipSDP in Julia with the COSMO solver which gave an accurate solution (identical to MOSEK) for small networks with SDP dim $10\times 10$. Unfortunately, for small MNIST networks of just two layers (resulting in an SDP of dim $800\times 800$), COSMO appears to be unstable and will not converge. Finally, we want to comment that COSMO aims to solve the KKT condition of conic programs, and does not guarantee the returned solution to be valid Lipschitz upper bounds.
> In contrast, our proposed first-order method LipDiff (EP-LipSDP)  guarantees that any value returned by LipDiff is always a valid upper bound higher than or equal to LipSDP’s optimal value, due to the fact that the iterations of LipDiff naturally correspond to feasible points of LipSDP which directly gives Lipschitz upper bounds. We have included the above discussion in Appendix G.6 of our revised paper.
>
>
>
>
> > "Dathathri et al. (2020) solved a different problem" => which problem?
>
> Verification of DNN subject to $\ell_\infty$ attack with radius $\epsilon$ (see [Raghunathan'18]).
>
> [Helmberg'00]Helmberg, Christoph, and Franz Rendl. A spectral bundle method for semidefinite programming. SIAM Journal on Optimization 10.3 (2000): 673-696
>
> [Liao'23]Feng-Yi Liao, Lijun Ding, and Yang Zheng. An overview and comparison of spectral bundle methods for primal and dual semidefinite programs. 2023
>
> [Ding'23]Lijun Ding and Benjamin Grimmer. Revisiting spectral bundle methods: Primal-dual (sub) linear convergence rates. SIAM Journal on Optimization, 2023
>
> [Raghunathan'18]Aditi Raghunathan, Jacob Steinhardt and Percy Liang. Semidefinite relaxations for certifying robustness to adversarial examples. NeurIPS 2018

---

### Official Review · Reviewer_LyYu · 2023-11-01

**Soundness:** 3 good
**Presentation:** 3 good
**Contribution:** 4 excellent
**Rating:** 6
**Confidence:** 3

**Summary:**

The authors tackle the problem of Lipschitz constant estimation of neural networks. Semidefinite programs (SDP) have shown great success for Lipschitz constant estimation, but the memory and time consumption avoid them to scale to modern neural network architectures. By transforming the problem into an eigenvalue problem, the proposed method is more memory efficient than the existing methods for solving SDP, and performs much better than simple matrix production norm bound.

**Strengths:**

1. The proposed method seems reasonable and promising.
2. The reduction into the eigenvalue problem is very useful and makes the proposed method practical and provides potential for further optimization.

**Weaknesses:**

1. While I can get that transforming into an eigenvalue problem would be beneficial (as there are many past developments in solving such problems), it is still kind of vague why and to what extent the proposed method is more memory efficient.

**Questions:**

1. Could the authors list the time and memory of each method, so it is clearer for comparison?
2. Is it possible to provide some kind of lower bound to show how good the proposed method is? LipSDP serves as a comparison target, but is only viable in one of the compared architectures. Showing that the proposed method is competitive with LipSDP on more architectures can make the paper more convincing.

---

> ### Author Response · Authors · 2023-11-22
> **Response to Reviewer LyYu**
>
> We appreciate your positive review. We address your concerns below.
>
> > While I can get that transforming into an eigenvalue problem would be beneficial (as there are many past developments in solving such problems), it is still kind of vague why and to what extent the proposed method is more memory efficient. Could the authors list the time and memory of each method, so it is clearer for comparison?
>
> Thanks for this useful comment.
> Now we compare the interior point method (IPM) with our proposed LipDiff method in a more transparent manner. The memory complexity and the per iteration computation compelxity of our first-order methods are much cheaper than IPM, while IPM requires less iterations in total to achieve an $\epsilon$-approximate solution ($\epsilon$ is the error in the final solution). To achieve an $\epsilon$-approximate solution, the total iteration number required by IPM is on the order of $O(\log(1/\epsilon))$. For our proposed LipDiff, we are applying a subgradient method to solve a convex nonsmooth optimization problem, and the total iteration number needed to get an $\epsilon$-approximate solution is on the order of $O(1/\epsilon^2)$  [Nesterov'03]. **However, for each iteration, the computational/memory efficiency for IPM is much worse than our first-order subgradient method.**
> Let $n$ be the number of nodes in the network, and $p$ be the number of parameters in the network. In general, $n< p< n^2$ and if the network is deep, $p<< n^2$. The per iteration memory complexity for IPM is $O(n^4)$ and computational complexity is $O(n^6)$ [Dathathri'20]. While for our method (LipDiff), the per iteration memory complexity is $O(L p)$, and per iteration computational complexity is $O(L p)$, where $L$ is the number of Lanczos iterations used for approximate the maximum eigenvalue (see Appendix G.5 of our revised paper for more details). In our evaluation, $L$ ranges from $15$ to $50$. In practice, Lipschitz estimation of neural networks typically does not require using very small $\epsilon$ value, i.e. $\epsilon$ is typically set to be on the order of $0.1$ or at most $0.01$. Therefore, we argue that per iteration complexity/efficiency matter more for the Lipschitz estimation problem of larger neural networks. In addition, we have developed the special analytical initialization (from Section 5 and Appendix B) to reduce the iteration number needed by LipDiff.
> As a matter of facts, IPM is not applicable to large networks on CIFAR, while our method can be used to improve the state-of-the-art Lipschitz upper bounds obtained using matrix norm product.
>  This justifies the significance of our contribution. We have added this discussion into Appendix G.5 of our revised paper.
>
>
>
>
>
> > Is it possible to provide some kind of lower bound to show how good the proposed method is? LipSDP serves as a comparison target, but is only viable in one of the compared architectures. Showing that the proposed method is competitive with LipSDP on more architectures can make the paper more convincing.
>
> We provide additional experiments in the general response section, with two additional architectures that mosek solver is able to handle and also a lower bound from sampling. We can see that LipDiff is quite competitive in comparison LipSDP on these extra experiments.
>
> [Dathathri'20]Sumanth Dathathri et.al. Enabling certification of verification-agnostic networks via memory-efficient semidefinite programming. NeurIPS 2020
>
> [Nesterov'03]Nesterov, Yurii. Introductory lectures on convex optimization: A basic course. Vol. 87. Springer Science & Business Media, 2003.

---

> > ### Comment · Reviewer_LyYu · 2023-11-23
> >
> > I thank the author for the response. I will maintain my score.

---

### Official Review · Reviewer_CCdS · 2023-11-13

**Soundness:** 4 excellent
**Presentation:** 3 good
**Contribution:** 4 excellent
**Rating:** 8
**Confidence:** 3

**Summary:**

The authors proposed an interesting reformulation (LipDiff) to use SDP to estimate the Lipschitz smoothness of a Neural Network (specifically, LipSDP.) The major observation is that there is a trace bound on the Shor's relaxation on the "Rayleigh quotient" QCQP of the Lipschitz-smoothness problem inspired by [Wang et al., 2022, section 5]. And inspired by [Liao et.al., 2023], the SDP primal problem with an explicit trace bound has an equivalent objective value to a dual optimization problem with only box constraints and a "simple" objective function with linear + $\rho\cdot$positive(eigmax(X)) for large enough penalty $\rho$ that's computable. For such a boxed-contrained optimization problem with maximum eigenvalue in the objective function, the authors apply the Lanczos-inspired optimization method [Dathathri et al., 2022] and try to exploit the sparse structure in LipDiff. In the experiment section, the authors demonstrate the proposed method on "classical" multiple-layer CNN trained on realistic datasets.

**Strengths:**

The paper is surprisingly correct, given so many omitted parts. The result that the LipSDP can be reformulated as a boxed-constrained minimax eigenvalue problem is surprising and probably has some deeper connections. Further, although the proposed method is "inspired" by many recent papers, there are generally some adaptions/improvements to the original formulation or proofs. For example, the QCQP "Rayleigh quotient" problem considered in  [Wang et al., 2022, section 5] is L-$\infty$-based, but it's L2-based in this work. The proofs in [Liao et al., 2023] require strict feasibility, but the authors generalized it to boxed constraints without requiring strict feasibility. The Lanczos-inspired optimization method in [Dathathri et al., 2022] doesn't exploit sparsity, but the authors try it in the work. Thus, the paper is not just a blind combination of various state-of-the-art methods. Thus, I would recommend an acceptance.

**Weaknesses:**

First, let me complain that the authors should have included more context in the description and derivations. This makes verification hard, but fortunately, the proofs are short and correct. Other than that, my biggest concern is the equivalence of the formulations in eq (1) and (3), in which the authors "proved in the footnote" by stating "it can be proved". Unfortunately, I didn't find the equivalence in the mentioned literature. In the worst case, eq (3) is an upper bound of eq (1) for the additional constraints (not equivalence), which is still okay due to its computational advantages. Further, the paper didn't do an approximation ratio analysis like [Wang et al., 2022, section 5] and other works. Finally, I am not sure the minimization of linear + positive(maxeig) objective function is easier to solve than the original SDP or the manifold alternatives (E.g., ManOpt.) The proposed formulation minimax in nature so the gradient-based method may be pretty unstable.

**Questions:**

1. Please explicitly clarify the relationship between the objective values in the different formulations (scaling / square roots).
2. P3 after eq (1): The "last" term, not the 2nd term.
3. Footnote 2: I'm unconvinced about the abovementioned equivalence. Also, please explain why it's $\epsilon/2$-smooth.
4. P5 on adding a trace norm: the rationale is not explained unless the reader goes to [Liao et.al., 2023].
5. P5 on the exact penalty: [Liao et al., 2023] doesn't apply directly to the non-negative constraints, but your proof is correct.
6. P13 in Appendix A: the $\Lambda$s are not defined. Also, why $\xi/2$?

---

> ### Author Response · Authors · 2023-11-22
> **Response to Reviewer CCdS, Part 1**
>
> Thanks for the valuable feedback. We address your comments below.
>
> > First, let me complain that the authors should have included more context in the description and derivations.
>
> Thanks for this useful comment. We have revised our paper to include more explanations and derivations. Specifically, we have added Appendix E to include detailed proofs showing the equivalence of (1), (3), and (6). We have explained the $\frac{\xi}{2}$-Lipschitzness in Remark 1 added to the end of Appendix E.1. Hopefully this made the paper more readable.
>
> > My biggest concern is the equivalence of the formulations in eq (1) and (3). In the worst case, eq (3) is only an upper bound of eq (1) for the additional constraints (not equivalence).
>
> Thanks for this comment. For completeness, we have added a proof to Appendix E.1 of our revised paper to show the equivalence of (1) and (3). To see why eq (3) provides a lower bound for (1), we can use a scaling argument. Let $(\hat{\zeta}^*,\hat{\tau}^*)$ be the optimal feasible point for (1). Then we can choose $\gamma=\sqrt{\hat{\zeta}^*}$, $\zeta=2\gamma=2\sqrt{\hat{\zeta}^*}$, and  $\tau=\gamma \hat{\tau}^*=\sqrt{\hat{\zeta}^*}\hat{\tau}^*$. The resultant choice of $(\zeta, \tau,\gamma)$ can be verified to be a feasible point for (3), and hence we know the optimal value of (3) is upper bounded by the value of $\zeta$ in this particular feasible point, which is $\zeta=2\sqrt{\hat{\zeta}^*}$. For details, please see the first half of the proof of Proposition 1 in Appendix E.1 of our revised paper.  The second half of the proof in Appendix E.1 shows the optimal value of (3) is lower bounded by $2\sqrt{\hat{\zeta}^*}$. Hence the optimal value of (3) is exactly equal to $2\sqrt{\hat{\zeta}^*}$. For completeness, we have added another proof showing the equivalence of (3) and (6) in Appendix E.2 of our revised paper.
>
>
> > Further, the paper didn't do an approximation ratio analysis like [Wang et al., 2022, section 5] and other works.
>
> The goal of our paper is to provide principled methods that can be used to solve LipSDP in a scalable and memory-efficient manner. We believe this is a self-contained problem, and the scope of our paper is quite reasonable for an ICLR submission.  We agree with the reviewer that extending the approximation ratio analysis for multi-layer network cases is an important future task, and this problem was posed as an open problem in [Wang et al., 2022]. We will pursue this direction in the future.

---

> ### Author Response · Authors · 2023-11-22
> **Response to Reviewer CCdS, Part 2**
>
> >  Finally, I am not sure the minimization of linear + positive(maxeig) objective function is easier to solve than the original SDP or the manifold alternatives (E.g., ManOpt.). The proposed formulation minimax in nature so the gradient-based method may be pretty unstable.
>
> Our resultant optimization is just a convex nonsmooth optimization which can be provably solved using subgradient methods or more advanced bundle methods. Notice that we are not treating our resultant optimization problem as a minimax problem. In contrast, we just treat the problem as a convex nonsmooth optimization problem (maxeig introduces the nonsmoothness). Subgradient methods are quite stable for such convex nonsmooth optimization problems. We discuss the difference between a convex nonsmooth optimization problem and a minimax problem in Appendix G.7 of our revised paper. To provide more insights into your question, suppose one is using interior point methods (IPM) to solve the original LipSDP, and we compare IPM with our first-order subgradient method LipDiff. The memory complexity and the per iteration computation complexity of our first-order methods are much cheaper than IPM, while IPM requires less iterations in total to achieve an $\epsilon$-approximate solution ($\epsilon$ is the error in the final solution). To achieve an $\epsilon$-approximate solution, the total iteration number required by IPM is on the order of $O(\log(1/\epsilon))$. For our proposed LipDiff, we are applying a subgradient method to solve a convex nonsmooth optimization problem, and the total iteration number needed to get an $\epsilon$-approximate solution is on the order of $O(1/\epsilon^2)$. **However, for each iteration, the computational/memory efficiency for IPM is much worse than our first-order subgradient method.**
> Let $n$ be the number of nodes in the network, and $p$ be the number of parameters in the network. In general, $n< p< n^2$ and if the network is deep, $p<< n^2$. The per iteration memory complexity for IPM is $O(n^4)$ and computational complexity is $O(n^6)$ [Dathathri'20]. In contrast, for our method (LipDiff), the per iteration memory complexity is $O(L p)$, and per iteration computational complexity is $O(L p)$, where $L$ is the number of Lanczos iterations used for approximate the maximum eigenvalue (see Appendix G.5 of our revised paper). In our evaluation, $L$ ranges from $15$ to $50$. In practice, Lipschitz estimation of neural networks typically does not require using very small $\epsilon$ value, i.e. $\epsilon$ is typically set to be on the order of $0.1$ or at most $0.01$. Therefore, we argue that per iteration complexity/efficiency matter more for the Lipschitz estimation problem of larger neural networks. Our analytical initialization also reduces the iteration number of LipDiff. We can see that IPM is not applicable to large networks on CIFAR, while our method can be used to improve the state-of-the-art Lipschitz upper bounds (i.e. matrix norm product). We added this discussion into Appendix G.5 of our revised paper.
>
> >  Please explicitly clarify the relationship between the objective values in the different formulations (scaling /square roots).
>
> We have formalized this relationship in Appendix E.1 of our revised paper, which is added to demonstrate the equivalence of (1) and (3). Specifically, denote the optimal values of (1) and (3) as $\hat{\zeta}^*$ and $\zeta^*$, respectively. We have $\zeta^*=2\sqrt{\hat{\zeta}^*}$. Please see Appendix E.1 for a detailed proof.
>
> > P3 after eq (1): The "last" term, not the 2nd term.
>
> Thanks for pointing this out. We have fixed this typo.
>
> > Footnote 2: I'm unconvinced about the abovementioned equivalence. Also, please explain why it's $\epsilon/2$-smooth
>
> We included details proofs for the equivalence of (1), (3), and (6) in Appendix E of our revised paper. We also provide an explanation of $\epsilon/2$-Lipschitz in Remark 1 added in the end of Appendix E.1.
>
> > P5 on adding a trace norm: the rationale is not explained unless the reader goes to [Liao et.al., 2023].
>
> Thanks for pointing this out. We have revised the end of the first paragraph of Section 4.1 to expose the rationale in [Liao'23] early.
>
> > P5 on the exact penalty
>
> Your understanding is correct.
>
> > P13 in Appendix A: the $\Lambda$s are not defined. Also, why  $\xi/2$?
>
> Thanks for pointing this out. We have $\Lambda_1'=diag(\tau')$, $\Lambda_2'=diag(\lambda')$, $\Lambda_1^0=diag(\tau^0)$, $\Lambda_2^0=diag(\lambda^0)$, $\Lambda_1=diag(\tau)$, and $\Lambda_2=diag(\lambda)$. We have added these definitions into Appendix A and Appendix B of our revised paper. The $\frac{\xi}{2}$-Lipschitzness is explained in Remark 1 added to Appendix E.1.
>
>
> [Dathathri'20] Enabling certification of verification-agnostic networks via memory-efficient semidefinite programming.
>
> [Liao'23] Feng-Yi Liao, Lijun Ding, and Yang Zheng. An overview and comparison of spectral bundle methods for primal and dual semidefinite programs.

---

### Author Response · Authors · 2023-11-22
**General Response**

We thank all reviewers for the professional review, which can significantly strengthen our paper. Since multiple reviewers are asking for additional numerical results, here we provide a general response documenting our additional experimental results. Then we will address each reviewers' comments individually. We have also incorporated all the reviewers' comments into our revised paper.

## Additional experiments
We conduct more experiments on two additional architectures that the LipSDP-mosek solver can handle. A single hidden layer with 512 hidden nodes (denoted as **WIDE**), and a two-hidden-layer network (denoted as **DNN3**), and each of the hidden layers has 512 ReLU nodes. Additionally, we provide two new baselines: **LB** and **LipSDP-SCS**. For LB, we randomly sampled 500,000 points from the input space and computed the maximum $\ell_2$ norm of all the gradients induced on these samples, which serve as a lower bound. For LipSDP-SCS, we used the default SCS solver provided by CVXPY to solve the SDP program, with max iterations 2,500. Notice that SCS is an ADMM-based **first-order** solver, similar to COSMO. The results are summarized in the following tables:

|     Lipschitz Constant       | Product | LipSDP-mosek | LipDiff-dense | LipDiff-ex | LipDiff | LipDiff-rand | LB   | LipSDP-SCS |
|------------|---------|--------------|---------|---------|---------|---------|------|------------|
| MNIST DNN  | 9.31    | 4.82         | 4.96    | 4.86    | 4.9     | 5.89    | 1.57 | 4.82       |
| MNIST WIDE | 9.27    | 4.87         | 4.93    | 4.89    | 4.93    | 5.57    | 1.51 | 4.87       |
| MNIST DNN3 | 17.29   | 7.43         | 7.6     | 7.51    | 7.76    | 11.47   | 2.03 | 7.38       |

| Time       | Product | LipSDP-mosek | LipDiff-dense | LipDiff-ex | LipDiff | LipDiff-rand | LB   | LipSDP-SCS |
|------------|---------|--------------|---------|---------|---------|---------|------|------------|
| MNIST DNN  | 0.13    | 54.57        | 12.48   | 19.27   | 28.69   | 29.27   | 4.86 | 2706.5     |
| MNIST WIDE | 0.1     | 214.67       | 29.03   | 41.85   | 65.89   | 64.1    | 4.88 | 5014.93    |
| MNIST DNN3 | 0.12    | 649.4        | 83.55   | 128.5   | 238.4   | 241.34  | 4.95 | 13736.16   |

| Memory     | Product | LipSDP-mosek | LipDiff-dense | LipDiff-ex | LipDiff | LipDiff-rand | LB   | LipSDP-SCS |
|------------|---------|--------------|---------|---------|---------|---------|------|------------|
| MNIST DNN  | 1.54    | 169.83       | 114     | 114     | 118     | 118     | 2134 | 256.46     |
| MNIST WIDE | 3.23    | 297.68       | 192     | 192     | 230     | 230     | 2424 | 415.9      |
| MNIST DNN3 | 3.23    | 585.6        | 284     | 284     | 362     | 362     | 2622 | 810.26     |

MNIST-DNN comes from our original evaluation. Because none of the upper bound tools (including LipSDP-SCS) except for LipDiff variants can scale to CNNs, we omit the experimental results here. Notice that because the evaluation models are small, LipSDP-dense achieves better running time than the sparse representation. We include the results in Appendix F of the revised paper.


Notice that we do not know how good the lower bound is because the number of samples is unlikely to be sufficient. For example, even if we only consider the vertices of the input on the [0,1]-hypercube, there would be $2^{784}$ vertices for MNIST and $2^{3072}$ vertices for CIFAR10, which are way greater than 500,000.


For SCS, we notice that if we set the max iterations too small (for example, 50), the result can be very unstable. It can return either 0 or inf, which is not useful at all. On MNIST-DNN3, SCS provided a value smaller than the MOSEK solver, which we do not know whether it is a valid upper bound for the Lipschitz constant. However, LipDiff always returns a valid upper bound.

Overall, we can see that our LipDiff method gives competitive results in comparison to the baselines.

---

### Meta-Review · Area_Chair_qTbn · 2023-12-18

**Metareview:**

The paper focuses on the problem of estimating the Lipshitz constant of neural networks. The main contribution is to transform the SDP-based formulations for Lipschitz constant estimation into an eigenvalue problem, which allows for a more efficient and scalable estimation. Almost all the reviewers (including myself) are in favor of accepting the paper. I recommend that the authors incorporate the excellent comments from the reviewers in the updated version (e.g. adding more details and context to the technical derivations in the main body, the additional experimental results, etc). The paper is a good contribution toward estimating the Lipshitz constant of neural networks.

**Justification For Why Not Higher Score:**

Some of the reviewers still had some (minor) concerns about the results, but the overall recommendation was to accept the paper.

**Justification For Why Not Lower Score:**

--

---

### Decision · Program_Chairs · 2024-01-16

Accept (poster)